psychology

state anxiety, trait anxiety, emotion, face, 7.5% carbon dioxide, bias

**Author for correspondence:**
Maddy L. Dyer
e-mail: maddy.dyer@bristol.ac.uk

# The role of state and trait anxiety in the processing of facial expressions of emotion

Maddy L. Dyer[1,2], Angela S. Attwood[1,2],
Ian S. Penton-Voak[1,3] and Marcus R. Munafò[1,2,3]

[1]School of Psychological Science, University of Bristol, Bristol, UK
[2]Medical Research Council Integrative Epidemiology Unit, University of Bristol, Bristol, UK
[3]National Institute for Health Research Bristol Biomedical Research Centre, University Hospitals Bristol NHS Foundation Trust, Bristol, UK

 MLD, 0000-0002-5924-4400; ASA, 0000-0003-3696-4349;
ISP, 0000-0003-4232-0953; MRM, 0000-0002-4049-993X

State anxiety appears to influence facial emotion processing (Attwood *et al.* 2017 *R. Soc. Open Sci.* **4**, 160855). We aimed to (i) replicate these findings and (ii) investigate the role of trait anxiety, in an experiment with healthy UK participants ($N = 48$, 50% male, 50% high trait anxiety). High and low state anxiety were induced via inhalations of 7.5% carbon dioxide enriched air and medical air, respectively. High state anxiety reduced global emotion recognition accuracy ($p = 0.01$, $\eta_p^2 = 0.14$), but it did not affect interpretation bias towards perceiving anger in ambiguous angry–happy facial morphs ($p = 0.18$, $\eta_p^2 = 0.04$). We found no clear evidence of a relationship between trait anxiety and global emotion recognition accuracy ($p = 0.60$, $\eta_p^2 = 0.01$) or interpretation bias towards perceiving anger ($p = 0.83$, $\eta_p^2 = 0.01$). However, there was greater interpretation bias towards perceiving anger (i.e. away from happiness) during heightened state anxiety, among individuals with high trait anxiety ($p = 0.03$, $dz = 0.33$). State anxiety appears to impair emotion recognition accuracy, and among individuals with high trait anxiety, it appears to increase biases towards perceiving anger (away from happiness). Trait anxiety alone does not appear to be associated with facial emotion processing.

## 1. Introduction

Facial expressions of emotion are, in part at least, innate and universal [1,2], and the ability to accurately recognize these non-verbal cues is vital for successful social interactions [3]. By signalling socially salient information such as fear, acceptance or threat [1], and thus an individual's behavioural intentions, facial expressions of emotion can activate approach or avoidance behaviour in the observer [4]. However, facial expressions of

emotion can be ambiguous, and they are prone to misinterpretation [5]. Deficits or biases in emotion processing, such as inaccurate emotion recognition or a tendency to perceive negativity in facial expressions, are associated with social, emotional and behavioural problems [6] and psychiatric disorders, including anxiety disorders [7].

Cognitive biases in emotion processing are common in anxiety [8]. People with anxiety disorders, who typically exhibit heightened trait and state anxiety, are characterized by processing biases towards emotionally threatening stimuli [9]. Cognitive models suggest that anxiety is related to cognitive biases at the stage of initial processing (i.e. attention). According to Gray's reinforcement sensitivity theory [10], the behavioural inhibition system promotes increased vigilance towards threat cues in anxiety. Preferential attention to and quick processing of real threat may confer a possible evolutionary advantage. For example, attention to facial expressions of negative emotions may have adaptive value by discouraging potentially costly interactions. However, excessive sensitivity to potential threat detection and its processing, which appears to operate in anxiety pathology, may be disadvantageous when an individual is biased to preferentially 'look' for feared stimuli. Indeed, hypervigilance towards and difficulty disengaging from threatening stimuli are thought to be central to the aetiology and maintenance of anxiety [11,12], as pharmacological interventions are associated with reductions in negative cognitive biases [13]. Therefore, a generally adaptive system of preferential processing of potential threats can become maladaptive when dysregulated (i.e. in anxiety disorders).

In support of these cognitive models, individuals with social anxiety disorder have been reported to display a greater sensitivity to anger [14], and they correctly identify anger [15] and disgust [16] at lower emotional intensities compared with non-anxious individuals. Anxiety severity is positively associated with recognition sensitivity for angry faces [17] and faster and more accurate recognition of fearful faces [18]. Furthermore, several studies suggest that individuals with trait anxiety show enhanced processing of fear. For example, trait anxiety is positively correlated with detection sensitivity for fearful faces [19,20] and individuals with high (versus low) trait anxiety show greater emotion recognition accuracy for fearful faces [21].

Conversely, other studies report deficits in facial emotion processing in anxiety. For example, one systematic review found a global deficit in emotion recognition among adults with anxiety disorders [22]. Other studies have found specific facial emotion processing impairments. For example, higher anxiety symptoms and anxiety disorders are associated with lower emotion recognition accuracy for anger expressions [18,23], patients with social anxiety disorder show reduced sensitivity in the recognition of anger and disgust [24], and individuals with generalized anxiety disorder show lower emotion recognition accuracy for sad expressions [25] compared with controls. Impairments in the speed of emotion recognition have also been noted. For example, individuals with social anxiety disorder display slower reaction times for recognizing surprise and happiness expressions [26,27]. Furthermore, there is evidence of negatively biased interpretation of neutral or ambiguous emotional facial expressions. For example, social anxiety is associated with greater tendencies to judge ambiguous facial expressions as angry (or other threatening emotions) [5,14,28]. High (versus low) trait anxious individuals more frequently erroneously categorize ambiguous [29] and neutral [30] facial expressions as fearful.

However, other studies have found no differences between socially anxious individuals and controls on emotion recognition accuracy [28,31], biased emotion recognition (neutral faces interpreted as negative) [26] or biased explicit detection of facial emotional expressions [32]. In addition, contrary to previous evidence, Cooper *et al.* [33] found no clear evidence of a difference in recognition accuracy of fearful facial expressions among individuals reporting high versus low trait anxiety.

Heterogeneity in anxiety could contribute to the inconsistent findings. Trait differences in anxiety exist between individuals and are more stable over time, whereas state variation in anxiety exists between individuals and within an individual over time [34,35]. State and trait anxiety are neuroanatomically and functionally distinct [36]. Furthermore, anxiety disorders are characterized by frequent, intense and persistent anxious states [37]. Differences in the measures of facial emotion processing between studies may also explain mixed findings. In short, emotion recognition accuracy may be measured by hit rate (e.g. the correct identification of anger if angry faces are presented). If an individual demonstrates a higher hit rate for anger, this suggests that they have superior emotion recognition accuracy for anger. However, this measure of emotion recognition accuracy does not account for the times when an individual identifies anger in faces that are not angry. These are known as false alarms/errors (i.e. the incorrect identification of the emotions presented). A bias towards making angry responses may manifest in a higher hit rate *and* a higher false alarm rate, whereas sensitivity reflects hit rate while accounting for false alarms. An additional complication is that the term 'bias' can also refer to different things (e.g. neutral or ambiguous emotional

facial expressions interpreted as angry) depending on the task and stimuli used in a study. Furthermore, different statistical analyses can be used to measure both bias and sensitivity.

There have been relatively few studies investigating the relationship between state anxiety and facial emotion processing. In one observational study, naturally occurring (i.e. not experimentally manipulated) state anxiety was correlated with reaction times to correctly identified fearful faces but not with detection sensitivity for fearful faces [20]. Previous studies have also used different methods to induce state anxiety experimentally. For example, aversive anticipation (threat of electric shock) increased recognition accuracy for fearful faces [30,38], compared with the non-threat condition. However, previous research from our group found contrary evidence, when using the 7.5% carbon dioxide ($CO_2$) challenge to induce state anxiety. This experimental manipulation compares the effects of a 20 min inhalation of 7.5% $CO_2$ enriched air versus a 20 min inhalation of medical air (control), while tasks are performed. Experimentally manipulated high state anxiety led to lower global emotion recognition accuracy (i.e. hits) and lower sensitivity (i.e. unbiased hit rate), compared with low state anxiety [39], consistent with observational analyses with naturally occurring state anxiety. These findings support cognitive models that argue that state anxiety impairs emotion processing [40]. The Clark & Wells [41] model of social phobia attributes this to a shift in attentional resources towards internal cues (i.e. anxiety symptoms) and away from external cues (i.e. facial expressions), which may also occur during the 7.5% $CO_2$ challenge. There was also evidence of increased interpretation biases during experimentally manipulated high (versus low) state anxiety (i.e. increased tendency to perceive anger and decreased tendency to perceive happiness in ambiguous angry–happy facial morphs) [39].

Williams *et al.* [42] distinguish between state and trait anxiety. According to their cognitive model, state anxiety influences the perception of threat (affective decision mechanism), and trait anxiety determines whether processing resources are directed towards (high trait anxiety) or away from (low trait anxiety) a stimulus perceived to be threatening (resource allocation mechanism). Therefore, high trait anxious individuals may have a greater attentional bias towards threat, whereas low trait anxious individuals may exhibit attentional avoidance. Research by MacLeod & Rutherford [9] supports this theory. They found that for individuals with high trait anxiety, state anxiety elicits a selective processing bias favouring threat-related information (colour naming of words). Whereas for individuals with low trait anxiety, state anxiety elicits a processing disadvantage for this threat-related information. However, there is a paucity of research investigating the interaction of state and trait anxiety on facial emotion processing. One study found no clear evidence that state social anxiety moderated the relationship between trait anxiety and sensitivity for fear, using an anxiety-provoking mood manipulation (pretend filming) [29]. Whereas another study found high state anxiety impaired emotion recognition more for people with high (versus low) trait anxiety [43]. Another investigated the interactive effects of sound-induced affective states and traits on emotion processing [34]; however, this was not an explicit measure of state anxiety. It is important to understand how both trait and state anxiety affect facial emotion processing individually, and in combination, as trait anxiety is characterized by more frequent and intense state anxious episodes, experienced across novel and everyday situations [35]. Dispositional differences in anxiety may, therefore, exacerbate state effects.

We conducted an experimental study using the 7.5% $CO_2$ challenge, a well-validated experimental model of state anxiety [44] which produces psychological and physiological symptoms [45–47]. Our primary aim was to replicate the main effect of state anxiety on facial emotion processing observed previously [39]. Our secondary aim was to investigate the association of trait anxiety with facial emotion processing and whether it moderates any effects of state anxiety. We hypothesized that (i) high state anxiety and (ii) high trait anxiety would lead to lower emotion recognition accuracy compared with low state and trait anxiety, respectively. We also hypothesized that (iii) high state anxiety and (iv) high trait anxiety would lead to increased interpretation bias towards perceiving anger (and decreased bias towards perceiving happiness), compared with low state and trait anxiety, respectively. In addition, we hypothesized that (v) the effects of high state anxiety on facial emotion processing would be greater among individuals who report high trait anxiety. State anxiety predictions were based on our previous findings, and trait anxiety predictions were based on the fact that high trait anxiety is characterized by increased frequency and intensity of state anxiety reactions compared with low trait anxiety [35], and previous studies show altered emotion processing in trait anxiety. Finally, as hypothesis-free secondary analyses, we explored the roles of state and trait anxiety on emotion recognition accuracy and sensitivity to specific emotions. Angry–happy facial morphs were selected to measure bias because cognitive models suggest biased threat detection in anxiety

[11], because previous stress-induction procedures have induced anger biases [48], and to ensure consistency with our previous experiment.

# 2. Methods

## 2.1. Design

We conducted a laboratory experiment with a mixed model design. Two computer tasks measured emotion recognition accuracy and interpretation bias towards perceiving anger (six-alternative forced choice (6AFC) and two-alternative forced choice (2AFC), respectively). For both emotion recognition accuracy and interpretation bias towards perceiving anger, there was a within-subjects factor of state anxiety (low, high), corresponding to medical air and 7.5% $CO_2$ enriched air conditions, respectively, and a between-subjects factor of trait anxiety (low, high). For analyses of emotion recognition accuracy, there was a second within-subjects factor of emotion (anger, disgust, fear, happiness, sadness and surprise). Gas inhalation order (air, 7.5% $CO_2$) and task order (2AFC, 6AFC) were counterbalanced across participants (http://www.randomizer.org). However, due to researcher error, gas order was unbalanced (23 participants air/$CO_2$, 25 participants $CO_2$/air). The study protocol was pre-registered on the Open Science Framework (https://osf.io/7y6qm/), and we obtained ethics approval from the Faculty of Science Human Research Ethics Committee at the University of Bristol (reference: 30041520644).

## 2.2. Participants

We recruited 48 participants (50% reporting high trait anxiety) from the local population (Bristol, UK) via mailing lists, poster and website adverts and by word of mouth. Participants were eligible if they met our criteria for low or high trait anxiety (details in the following paragraph), were aged between 18 and 50 years and were healthy. Exclusion criteria included recent alcohol and illicit drug use, daily smoking, high alcohol and caffeine consumption, pregnancy, abnormal blood pressure (BP) and/or heart rate (HR) and significant current or past physical or psychological illness. The study protocol (https://osf.io/7y6qm/) provides a complete, operationalized list of eligibility criteria.

To determine the trait anxiety cut-off scores on the Spielberger State-Trait Anxiety Inventory (Trait Anxiety Subscale; STAI-trait) [35], we used data from our previous 7.5% $CO_2$ inhalation studies (comprising 15 studies and 425 participants). After excluding top-end outliers ($n = 8$), we split the remaining distribution into tertiles, which produced an upper limit for our low anxious group of 31 and a lower limit of our high anxious group of 44 (inclusive). Participants who scored within the middle tertile [32–43] were not recruited. For safety reasons, we applied an upper cut-off of 64 for the high trait anxiety group, as high trait anxiety is commonly associated with clinical anxiety [37]. This was the upper score we obtained in our full cohort (after exclusion of one outlier). Scores between 21–31 and 41–64 denoted low and high trait anxiety, respectively.[1]

Target sample size was calculated from a previous study in our group, which compared global emotion recognition accuracy (i.e. global hits count) in response to low state anxiety ($M = 134$, s.d. = 14) and high state anxiety ($M = 123$, s.d. = 19) conditions ($n = 44$; Cohen's $dz = 0.69$) [39]. Based on these data, we required a sample size of 24 participants to achieve 90% power to replicate this main effect, at an alpha level of 5%. We had no data from which to estimate the magnitude of any state × trait anxiety interactions. To increase the power to detect an interaction effect, we recruited 48 participants, assuming Cohen's $dz = 0.69$.

## 2.3. Material and measures

### 2.3.1. Gas mixtures

The medical air cylinders contained 21% $O_2$ (low state anxiety condition), and the 7.5% $CO_2$ cylinders contained 7.5% $CO_2$/21% oxygen/71.5% nitrogen (high state anxiety condition) (BOC Ltd, UK). Gases

---

[1]Scores between 21–31 and 44–64 denoted low and high trait anxiety, respectively, for the majority of participants. For the last two participants, we lowered the threshold of inclusion for high trait anxiety to 41–64 to support recruitment.

|  | anger | disgust | fear | happiness | sadness | surprise |
|---|---|---|---|---|---|---|

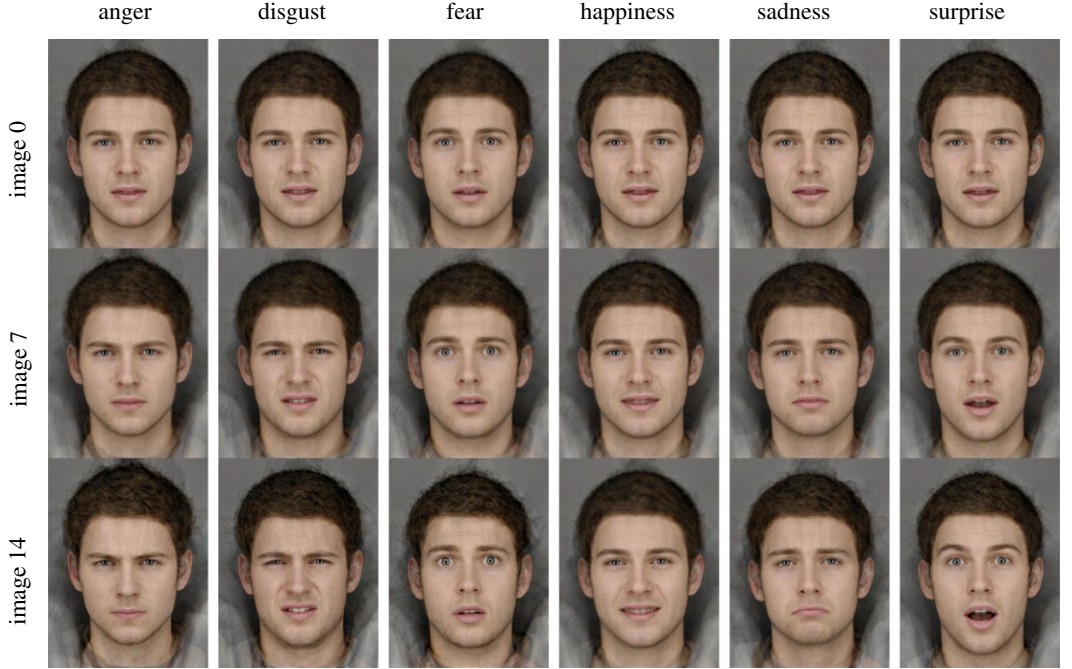

**Figure 1.** Selection of stimuli from the six-alternative forced choice (6AFC) task which measures emotion recognition accuracy. Note: image 0 = the emotional prototype (5% emotional signal); image 14 = the emotional exemplar (100% emotional signal).

were administered using an oro-nasal mask (Hans Rudolph Inc., USA), and this was single blind for safety reasons.

### 2.3.2. Questionnaires

State and trait anxiety were assessed using the STAI-state and STAI-trait [35]. Positive and negative affect were determined using the positive and negative affect schedule (PANAS) [49] and visual analogue scales (VAS). The VAS had 11 items (alert, sedated, fearful, relaxed, anxious, happy, feel like leaving, tense, nervous, worried and stressed) from 0 (not at all) to 100 (extremely). The anxiety sensitivity index (ASI) [50] and the Eysenck personality questionnaire—revised (EPQ-R; extraversion, neuroticism, psychoticism and lie factors) [51] described the sample. The questionnaires are reliable and valid measures of the constructs they were intended to assess [52–55].

### 2.3.3. Six-alternative forced choice (6AFC) task

The 6AFC stimuli were composite images (digital averages) of the six basic facial expressions of emotion (anger, disgust, fear, happiness, sadness and surprise). These were originally created for the Cambridge Cognition Emotion Recognition Task in the following steps. First, six 'emotional exemplars' were created by combining shape and colour information from photographs of 12 young adult white males expressing each emotion. For photography details, see Eastwood *et al.* [56]. Second, one 'emotional prototype' face was created by averaging the six emotional exemplars. There is evidence that an emotional prototype face is likely to be a better approximation of the centre of emotional 'face space' than a neutral face [57], and a neutral face is not without emotion. For example, neutral and angry faces have been found to elicit comparable negative facial responses when passively viewed, which may indicate that neutral faces are perceived to be negatively valenced [58]. Finally, 15-image linear morph sequences were generated for each emotion, ranging in equally spaced emotional intensities from the emotional prototype (an emotionally ambiguous face; 5% emotional signal) to the emotional exemplar (the full intensity unambiguous emotion; 100% emotional signal). Each image between the emotional prototype and the emotional exemplar contained a proportion of both images (e.g. 90% happiness contained 10% emotional prototype). There were 90 images in total (figure 1).

The 6AFC task contained two blocks of 90 trials each. Trials began with a fixation point (1500–2500 ms). Participants were presented with a face image for 150 ms, followed by a backwards mask (visual white noise) of 250 ms to avoid processing of after-images. The next screen displayed six

happy ⟷ angry

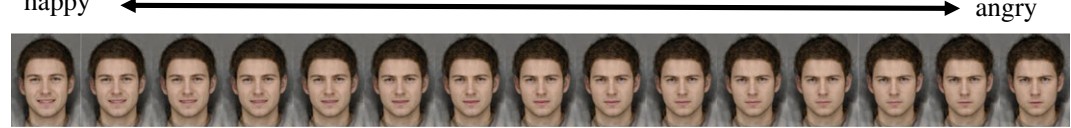

**Figure 2.** Stimuli from the two-alternative forced choice (2AFC) task, which measures interpretation bias towards perceiving anger.

emotional descriptors (anger, disgust, fear, happiness, sadness and surprise) in a circular arrangement. Participants were required to select the descriptor that best described the emotion presented, as quickly and accurately as possible using the mouse. Descriptors remained on screen for 10 s or until the participant responded. A fixation cross was displayed between images (between 500 and 1500 ms). Each of the 90 images (15 per emotion) were presented twice at random, and the task lasted approximately 12 min.

The primary outcomes from the 6AFC were the number of correct identifications for each emotion (i.e. hits; emotion recognition accuracy), and the proportion of incorrect identifications (i.e. false alarms) for each emotion. As described below in the statistical analysis section, false alarms were not analysed separately. Instead, hit rate and false alarm rate for each emotion were used to compute sensitivity scores (i.e. A-prime).

### 2.3.4. Two-alternative forced choice (2AFC) task

The 2AFC stimuli were 15-image linear morph sequences ranging in equally spaced emotional intensities from the full intensity happy exemplar to the full intensity angry exemplar from the 6AFC (figure 2). Each image between the two full intensity images contained a proportion of both emotions (e.g. 90% happiness contained 10% anger). The central images in the sequences were emotionally ambiguous. The 2AFC task contained 45 trials. Trials began with a fixation point (between 1500 and 2500 ms at random). Participants were presented with a face image for 150 ms, followed by a backwards mask (visual white noise) of 250 ms. A response prompt appeared until a response was made. Participants decided whether the emotion presented was 'happy' or 'angry', by pressing designated keys as quickly and accurately as possible. Each image was presented three times at random, and the task lasted approximately 3 min.

The primary outcome from the 2AFC was interpretation bias towards perceiving anger. This score was calculated as a 'balance point' estimate (threshold score) on the continuum at which the participant was equally likely to perceive happiness or anger. The balance point was estimated by calculating the number of 'happy' responses divided by the number of trials (i.e. 45), multiplied by the number of stimuli (i.e. 15). As the continuum ranged from happy (image 1) to angry (image 15), lower threshold scores indicated greater interpretation biases towards perceiving anger (i.e. individuals show an earlier change from perceiving happiness to anger) relative to higher threshold scores. In other words, relatively more morphed faces on the continuum (that contain a proportion of happiness and anger) are perceived to be angry than happy. The computer tasks were created and run using E-prime (Psychology Software Tools Inc., USA).

### 2.4. Procedure

After reading the study information sheet (provided online or via email), interested participants received a link to an online survey that screened for high and low trait anxiety. Participants who were eligible based these scores were contacted to complete a brief telephone screening. This included questions on age, gender, caffeine and alcohol consumption, smoking habits, brief medical history and general practitioner details.

On arrival at the test session, participants provided written informed consent and were reminded of their right to withdraw during the study. Participants were objectively screened for smoking (piCO Smokerlyzer), recent alcohol consumption (AlcoDigital 3000 breathalyzer), unhealthy body mass index, unhealthy BP and HR (OMRON M6 BP monitor) and pregnancy and recent drug use (urine screen). All other criteria, including psychiatric health (using a screening tool adapted from the Mini-International Neuropsychiatric Interview) [59], were assessed by self-report.

Enrolled participants completed baseline questionnaires (STAI-state, PANAS, ASI, VAS), and HR and BP were measured. Participants were fitted with an oro-nasal mask. They inhaled the gas (air or 7.5%

$CO_2$) for 1 min to allow anxiety levels to stabilize and then they completed both computer tasks (6AFC, 2AFC) while continuing to inhale the gas. Inhalations lasted up to 20 min. Immediately after the inhalation, masks were removed, HR and BP were measured again and participants completed the STAI-state, PANAS, VAS (reporting on how they felt during the inhalation when the effects were at their peak). There was a 30 min rest period between inhalations to minimize carry-over effects. The second inhalation followed the identical procedure. Finally, participants completed the EPQ-R and then remained in the laboratory for 20 min to ensure good recovery. Participants received a written debrief sheet and were reimbursed £20. The session lasted approximately 2.5 h. Participants were also given a safety card, and a follow-up call was made 24 h later to record any adverse effects.

## 2.5. Statistical analysis

Data were analysed using IBM SPSS Statistics (v. 25). To check the internal validity of the state anxiety experimental manipulation, we compared self-reported (STAI-state, PANAS, VAS) and physiological (BP, HR) outcomes after high state anxiety (7.5% $CO_2$) versus low state anxiety (air) conditions, using paired sample $t$-tests. We checked interpretation bias towards perceiving anger and emotion recognition accuracy (global hits) data for outliers using boxplots. Outliers were defined as scores over 1.5 times the interquartile range (IQR) above the upper or below the lower quartile in both the low and high state anxiety conditions, or three times the IQR above the upper or below the lower quartile in one condition. Normality (skewness and kurtosis), homogeneity of variance (Levene's test) and sphericity (Mauchly's test) were checked. Where the assumption of sphericity was violated, Greenhouse Geisser statistics are reported.

For emotion recognition accuracy, the primary statistical model was a 2 × 2 mixed model analysis of variance (ANOVA) with state anxiety (low, high) as the within-subjects factor and trait anxiety (low, high) as the between-subjects factor. Emotion recognition accuracy, as measured by hits, was the primary dependent variable from the 6AFC task. The secondary statistical model for emotion recognition accuracy was a 2 × 2 × 6 mixed model ANOVA with an additional within-subjects factor of emotion (anger, disgust, fear, happiness, sadness, surprise). The state anxiety × emotion interaction explored the emotion-specificity of state anxiety effects on emotion recognition accuracy, and the full model explored whether trait anxiety moderated this effect. For interpretation bias towards perceiving anger, the primary statistical model was a 2 × 2 mixed model ANOVA with state anxiety (low, high) as the within-subjects factor and trait anxiety (low, high) as the between-subjects factor. Interpretation bias towards perceiving anger (away from happiness), as measured by threshold scores, was the primary dependent variable from the 2AFC task. Where there was evidence of an interaction, we conducted *post hoc* simple effects analyses ($t$-tests). Results are framed in terms of the strength of evidence against the null hypothesis (e.g. $p < 0.05$ provides modest evidence while $p < 0.001$ provides strong evidence) [60]. Cohen [61] has also provided conventions to define small ($\eta_p^2 = 0.01$; $dz = 0.20$), medium ($\eta_p^2 = 0.06$; $dz = 0.50$) and large ($\eta_p^2 = 0.14$; $dz = 0.80$) effects.

In subsequent analyses, we deviated from our pre-registered protocol in three ways. First, we planned to analyse false alarm and sensitivity outcome data (from the 6AFC task) using the same statistical models described for hits. However, as stated previously [39], the proposed 2 (state anxiety) × 2 (trait anxiety) models were later considered to be unsuitable, as false alarms and sensitivity scores are only meaningful when considered at an emotion-specific level. Since emotion-specific false alarms and thus emotion-specific sensitivity scores are not independent (i.e. changes in false alarm rate for one emotion will affect false alarm rates for the other five emotions), the proposed 2 (state anxiety) × 2 (trait anxiety) × 6 (emotion) models for false alarm and sensitivity outcome data violated an ANOVA assumption. Therefore, we conducted six separate 2 (state anxiety) × 2 (trait anxiety) models to examine effects on emotion-specific sensitivity. Second, we decided not to analyse false alarm data separately, as they are less informative than sensitivity scores. Sensitivity measures the degree of overlap between signal (presence of target emotion) and noise (absence of target emotion) distributions [62]. It, therefore, indicates whether there is a genuine improvement in recognizing a specific emotion, rather than a general tendency to identify that emotion regardless of whether it is present (i.e. bias). Finally, based on recent research using these tasks [56], we decided to use a different measure of sensitivity (A-prime; A′) to the one used previously (unbiased hit rate). Unbiased hit rate is 'the joint probability that a stimulus category is correctly identified given that it is presented at all and that a response is correctly used given that it is used at all' [63]. It is a measure of perceptual sensitivity which weights hit rate by false alarm rate. A′ is a distribution-free, non-parametric, signal detection measure of sensitivity [62]. We calculated sensitivity scores from hit rate

**Table 1.** Differences in participant baseline characteristics between trait anxiety groups and differences in state anxiety measures following 7.5% carbon dioxide and air inhalations. Note: $N = 48$. Paired-samples $t$-tests (Cohen's $dz$ effect sizes) for comparisons between inhalations, and independent samples $t$-tests (Cohen's $d$ effect sizes) for comparisons between trait anxiety groups. STAI-state, State-Trait Anxiety Inventory State Subscale; PANAS, positive and negative affect schedule; VAS, visual analogue scale; SBP, systolic blood pressure; DBP, diastolic blood pressure; HR, heart rate; s.e., standard error; CI, confidence interval.

| measure | mean difference (s.e.) | $t$ | 95% CI | $p$-value | effect size |
|---|---|---|---|---|---|
| high versus low trait anxiety | | | | | |
| age | 1.25 (0.70) | 1.80 | −0.15 to 2.65 | 0.08 | 0.52 |
| anxiety sensitivity | 5.0 (2.18) | 2.29 | 0.61–9.39 | 0.03 | 0.66 |
| extraversion | −3.42 (1.25) | −2.74 | −5.94 to −0.89 | 0.01 | 0.79 |
| neuroticism | 7.17 (1.33) | 5.40 | 4.49–9.84 | <0.001 | 1.56 |
| psychoticism | 0.67 (0.96) | 0.69 | −1.27 to 2.60 | 0.49 | 0.20 |
| lie | −0.29 (1.21) | −0.24 | −2.72 to 2.14 | 0.81 | 0.07 |
| 7.5% carbon dioxide (high state anxiety) versus air (low state anxiety) | | | | | |
| STAI-state | 13.15 (1.42) | 9.28 | 10.30–16.00 | <0.001 | 1.34 |
| PANAS-positive | −2.23 (0.81) | −2.74 | −3.87 to −0.59 | 0.01 | 0.40 |
| PANAS-negative | 5.04 (0.69) | 7.36 | 3.66–6.42 | <0.001 | 1.06 |
| VAS-positive | −14.01 (3.01) | −4.65 | −20.07 to −7.95 | <0.001 | 0.67 |
| VAS-negative | 18.62 (2.67) | 6.96 | 13.24–24.00 | <0.001 | 1.00 |
| SBP | 7.31 (1.56) | 4.70 | 4.18–10.44 | <0.001 | 0.68 |
| DBP | −0.75 (1.02) | −0.74 | −2.79 to 1.29 | 0.46 | 0.11 |
| HR | 14.54 (2.15) | 6.76 | 10.21–18.87 | <0.001 | 0.98 |

and false alarm rate data for each emotion in each state anxiety condition, based on the formula provided by Fisk *et al.* [64]. For a small number of cases where hits were less than false alarms, we used an alternative formula [62], verified using software [65]. A′ sensitivity scores typically range from 0.5 (signal cannot be distinguished from noise) to 1 (perfect performance), values less than 0.5 may arise from sampling error or response confusion, and the minimum possible value is 0 [62].

# 3. Results

Data are available at the University of Bristol data repository, data.bris, at https://doi.org/10.5523/bris. 31ddg7ihazewo2iqggjowqlt7g.

## 3.1. Participant characteristics

We recruited 48 participants, but only 45 had complete outcome data due to a technological fault which meant task data were not recorded. One participant had 2AFC task data only, one participant had 6AFC task data only and one participant had no outcome data. We, therefore, had 87% power to detect our target effect size of $dz = 0.69$. Participants with data on at least one outcome ($n = 47$, 51% male) were aged between 18 and 29 years ($M = 21.5$, s.d. = 2.5). Trait anxiety and anxiety sensitivity scores ranged from 21–57 ($M = 38.8$, s.d. = 12.0) to 1–41 ($M = 14.6$, s.d. = 7.9), respectively. EPQ-R scores ranged from 5–23 ($M = 15.7$, s.d. = 4.6) for extraversion, 0–22 ($M = 9.3$, s.d. = 5.9) for neuroticism, 1–14 ($M = 6.9$, s.d. = 3.3) for psychoticism to 0–19 ($M = 8.3$, s.d. = 4.0) for the lie subscale. Participant characteristics between groups were similar, except that anxiety sensitivity and neuroticism were higher and extraversion was lower in the high (versus low) trait anxiety group (table 1).

## 3.2. Manipulation check

State anxiety, negative affect (PANAS-negative, VAS-negative), systolic BP and HR were higher, and positive affect (PANAS-positive, VAS-positive) was lower, after the high state anxiety condition (7.5%

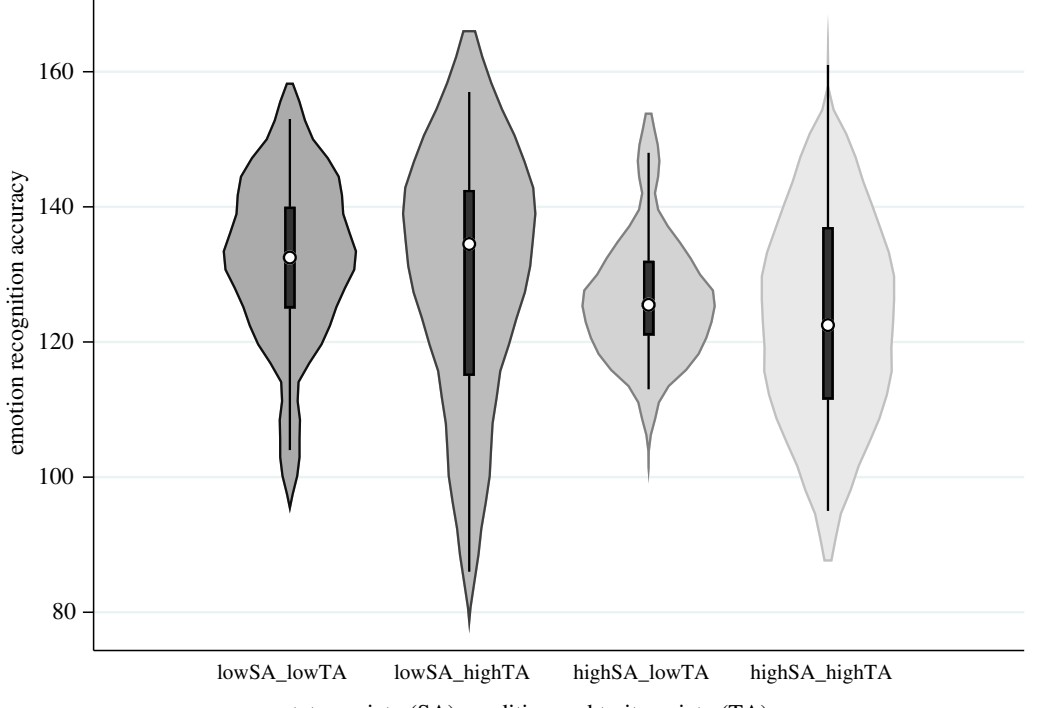

**Figure 3.** The role of state and trait anxiety on emotion recognition accuracy. Note: violin plots show the median (circle), boxplot and kernel density estimation of global hits (total) on the six-alternative forced choice (6AFC) task, which represent emotion recognition accuracy, for each condition. SA, state anxiety; TA, trait anxiety.

**Table 2.** Differences in hits (recognition accuracy) for each emotion, in the high state anxiety and low state anxiety conditions. Note: $N = 46$. CI, confidence interval.

| emotion | mean difference (s.d.): high versus low state anxiety | $t$ | 95% CI | $p$-value | Cohen's $dz$ |
|---|---|---|---|---|---|
| anger | −0.09 (2.87) | −0.21 | −0.94 to 0.77 | 0.84 | 0.03 |
| disgust | −0.93 (3.17) | −2.00 | −1.87 to 0.01 | 0.05 | 0.29 |
| fear | −1.85 (5.99) | −2.09 | −3.63 to −0.07 | 0.04 | 0.31 |
| happiness | −2.50 (5.07) | −3.35 | −4.00 to −1.00 | 0.002 | 0.49 |
| sadness | 0.48 (3.32) | 0.98 | −0.51 to 1.46 | 0.33 | 0.14 |
| surprise | −0.07 (3.14) | −0.14 | −1.00 to 0.87 | 0.89 | 0.02 |

$CO_2$) than the low state anxiety condition (air), which confirmed the internal validity of our state anxiety experimental manipulation (table 1). However, there was no clear evidence of a difference in diastolic BP.

## 3.3. Emotion recognition accuracy

In the 2 (state anxiety) × 2 (trait anxiety) model, there was evidence of a main effect of state anxiety on global emotion recognition accuracy [$F_{1,44} = 6.89$, $p = 0.01$, $\eta_p^2 = 0.14$]. Accuracy was lower in the high ($M = 125.11$, s.d. = 13.60) versus the low ($M = 130.07$, s.d. = 16.85) state anxiety condition. There was no clear evidence of a main effect of trait anxiety [$F_{1,44} = 0.28$, $p = 0.60$, $\eta_p^2 = 0.01$] or a state anxiety × trait anxiety interaction [$F_{1,44} = 0.07$, $p = 0.80$, $\eta_p^2 = 0.001$] on global emotion recognition accuracy (figure 3).

In the 2 (state anxiety) × 2 (trait anxiety) × 6 (emotion) model, there was strong evidence of a main effect of emotion [$F_{3.16,138.91} = 23.48$, $p = <0.001$, $\eta_p^2 = 0.35$] on emotion recognition accuracy. There was also evidence of a state anxiety × emotion interaction [$F_{3.76,165.36} = 4.11$, $p = 0.004$, $\eta_p^2 = 0.09$]. Greenhouse Geisser corrections were applied to both effects. *Post hoc* paired sample $t$-tests indicated lower accuracy in the high (versus low) state anxiety condition for disgust, fear and particularly happiness (table 2). There was no clear evidence of a difference in accuracy between conditions for anger, sadness and surprise. There was no clear evidence of

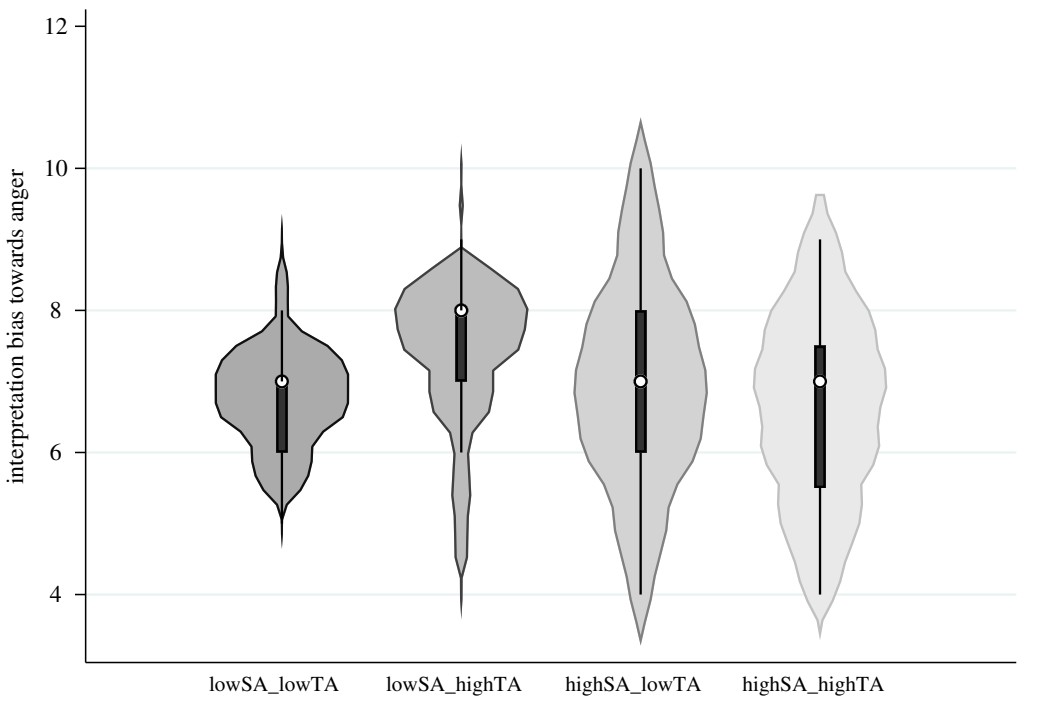

**Figure 4.** The role of state and trait anxiety on interpretation bias towards perceiving anger (away from happiness). Note: violin plots show the median (circle), boxplot and kernel density estimation of threshold scores on the two-alternative forced choice (2AFC) task, which represent interpretation bias towards perceiving anger (away from happiness), for each condition. SA, state anxiety; TA, trait anxiety.

a trait anxiety × emotion interaction [$F_{3.16,138.91} = 0.65$, $p = 0.59$, $\eta_p^2 = 0.02$] or a state anxiety × trait anxiety × emotion interaction [$F_{3.76,165.36} = 0.57$, $p = 0.67$, $\eta_p^2 = 0.01$].

## 3.4. Interpretation bias towards perceiving anger

There was no clear evidence of a main effect of state anxiety on interpretation bias towards perceiving anger (away from happiness) [$F_{1,44} = 1.87$, $p = 0.18$, $\eta_p^2 = 0.04$]. There was no clear evidence of a main effect of trait anxiety [$F_{1,44} = 0.05$, $p = 0.83$, $\eta_p^2 = 0.001$]. However, there was modest evidence of a state anxiety × trait anxiety interaction ($F_{1,44} = 4.29$, $p = 0.04$, $\eta_p^2 = 0.09$) (figure 4). In the high trait anxiety group, there was some evidence of a difference in interpretation bias towards perceiving anger between the low and high state anxiety conditions (7.29 versus 6.63, $p = 0.03$, $dz = 0.33$), indicating greater bias towards perceiving anger when experiencing high state anxiety, but there was no evidence in the low trait anxiety group (6.82 versus 6.95, $p = 0.59$, $dz = 0.08$). In the low and high state anxiety conditions, there was no clear evidence of a difference in interpretation bias towards perceiving anger between individuals who report low versus high trait anxiety (6.82 versus 7.29, $p = 0.16$, $dz = 0.20$; 6.95 versus 6.63, $p = 0.44$, $dz = 0.11$, respectively).

## 3.5. Emotion recognition sensitivity

Descriptive statistics for sensitivity and accuracy (hit rate), separated by trait anxiety and emotion categories, are shown in table 3. There was evidence for a main effect of state anxiety on recognition sensitivity to happiness [$F_{1,44} = 9.20$, $p = 0.004$, $\eta_p^2 = 0.17$], with lower scores in the high ($M = 0.90$, s.d. = 0.04) than the low ($M = 0.92$, s.d. = 0.04) state anxiety condition. There was no clear evidence of a main effect of state or trait anxiety (or interaction) on recognition sensitivity for any other emotion ($p$s > 0.10; electronic supplementary material, table S1).

## 3.6. Order effects

There was no clear evidence that gas inhalation order or task order modified the effects of state and trait anxiety on interpretation bias towards perceiving anger ($p$s > 0.25). However, gas inhalation order modified the effect

**Table 3.** Mean (s.d.) hit rates (recognition accuracy) and sensitivity scores (A′) for each emotion, by trait anxiety group and state anxiety condition. Note: $N = 46$. Low state anxiety = air inhalation; high state anxiety = 7.5% $CO_2$ inhalation; (A′) = A-prime.

| outcome | emotion | trait anxiety | low state anxiety | high state anxiety |
|---|---|---|---|---|
| hit rate | anger | low | 0.63 (0.13) | 0.64 (0.10) |
| | | high | 0.66 (0.17) | 0.65 (0.15) |
| | disgust | low | 0.81 (0.09) | 0.77 (0.09) |
| | | high | 0.78 (0.19) | 0.76 (0.16) |
| | fear | low | 0.61 (0.24) | 0.53 (0.22) |
| | | high | 0.54 (0.30) | 0.50 (0.25) |
| | happiness | low | 0.79 (0.14) | 0.73 (0.18) |
| | | high | 0.76 (0.13) | 0.66 (0.21) |
| | sadness | low | 0.79 (0.10) | 0.81 (0.11) |
| | | high | 0.81 (0.13) | 0.83 (0.12) |
| | surprise | low | 0.73 (0.09) | 0.73 (0.12) |
| | | high | 0.75 (0.10) | 0.74 (0.11) |
| sensitivity | anger | low | 0.90 (0.04) | 0.90 (0.03) |
| | | high | 0.90 (0.05) | 0.90 (0.04) |
| | disgust | low | 0.93 (0.03) | 0.92 (0.03) |
| | | high | 0.93 (0.06) | 0.92 (0.05) |
| | fear | low | 0.86 (0.14) | 0.83 (0.14) |
| | | high | 0.80 (0.22) | 0.81 (0.15) |
| | happiness | low | 0.92 (0.04) | 0.91 (0.04) |
| | | high | 0.92 (0.03) | 0.90 (0.05) |
| | sadness | low | 0.92 (0.02) | 0.93 (0.03) |
| | | high | 0.93 (0.04) | 0.92 (0.03) |
| | surprise | low | 0.90 (0.03) | 0.90 (0.04) |
| | | high | 0.90 (0.04) | 0.89 (0.04) |

of state anxiety on emotion recognition accuracy ($p = 0.01$), with stronger effects when the 7.5% $CO_2$ inhalation came first (electronic supplementary material, table S2). This is a common finding in $CO_2$ inhalation studies, reflecting a possible effect of anticipatory anxiety in addition to the $CO_2$ induced anxiety.

## 3.7. Internal consistency

Finally, we conducted *post hoc* internal consistency analyses of the measurements obtained from the questionnaires and tasks (electronic supplementary material, table S3). McDonald's omega [66] was used for the questionnaires, Cronbach's alpha was used for the 6AFC task and the parallel forms reliability method was used for the 2AFC task because alternative methods (e.g. Cronbach's alpha) could not be applied to the 2AFC task (see the electronic supplementary material, for details). As shown in electronic supplementary material, table S3, the measurements obtained from the questionnaires and the 6AFC task (emotion recognition accuracy) had high internal consistency reliability, whereas the measurements obtained from the 2AFC task (interpretation bias towards perceiving anger) had moderate internal consistency reliability.

# 4. Discussion

We aimed to replicate the main effects of state anxiety on facial emotion processing (emotion recognition accuracy and interpretation bias towards perceiving anger) observed previously [39]. We also aimed to

extend these findings by investigating the association of trait anxiety with facial emotion processing, and whether trait anxiety moderates any effects of state anxiety. In support of our first hypothesis and the previous study, high (versus low) state anxiety led to lower global emotion recognition accuracy. This is consistent with a previous systematic review which found a global deficit in emotion recognition accuracy among individuals with anxiety disorders [22], who experience more frequent and intense anxious states. Contrary to our second hypothesis, we did not find evidence to indicate a relationship between trait anxiety and emotion recognition accuracy. Furthermore, trait anxiety did not appear to moderate effects of state anxiety on emotion recognition accuracy, contrary to predictions and some previous evidence [43].

Contrary to our third hypothesis and the previous study, interpretation bias towards perceiving anger did not appear to differ under high and low state anxiety conditions. Furthermore, we did not find evidence to indicate a relationship between trait anxiety and interpretation bias towards perceiving anger. Therefore, these findings are not consistent with cognitive theories which suggest greater attentional bias towards threat in anxiety [10,42] or previous evidence of an anger emotion processing bias in anxiety [5,14,28]. *Post hoc* analyses suggest that the measurements obtained from the 2AFC task (interpretation bias towards perceiving anger) had moderate reliability as measured by internal consistency. As stated by Parsons *et al.* [67], 'reliability is estimated from the scores obtained with a particular task performed by a particular sample under specific circumstances'. Therefore, although the task and the method of anxiety induction were consistent across the current and previous study, it is possible that the samples differed in some way, which may account for this lack of replication. Nonetheless, there are no *a priori* reasons to suspect that this sample is systematically different from the previous study, and so the robustness of our earlier finding is questionable.

However, as predicted, there was some evidence of an interaction between state and trait anxiety on interpretation bias towards perceiving anger. *Post hoc* tests indicated an increased tendency to perceive anger and a decreased tendency to perceive happiness in the high (versus low) state anxiety condition, among individuals with high (but not low) trait anxiety. In other words, a situational spike in anxiety appeared to cause a greater anger bias in emotion processing for individuals with a dispositional tendency to experience anxiety. These findings support cognitive models which propose that different patterns of bias in high and low trait anxious individuals become more pronounced as stimulus threat value or state anxiety increases [42]. Although results should still be interpreted with caution, only angry–happy facial morphs were included.

We also conducted hypothesis-free secondary analyses of emotion recognition accuracy data with an additional within-subjects factor of emotion, which we applied to hits. We found strong evidence for a main effect of emotion, and a state anxiety × emotion interaction (although the effect size was smaller) on emotion recognition accuracy, supporting the previous study. *Post hoc* tests indicated that high state anxiety led to lower emotion recognition accuracy for happiness, disgust and fear. However, there was no clear evidence of a difference in emotion recognition accuracy between high and low state anxiety conditions for angry, sad and surprised facial expressions. These findings largely support the previous study, although Attwood *et al.* [39] observed a deficit in recognition accuracy for anger (Study 1) and surprise (Study 2), in addition to happiness, disgust and fear. Discrepancies like these are common in this field. However, because we have used similar methods in both studies, this suggests that the effects may be transient or possible sample differences may explain this discrepancy. Interestingly, across all three studies, the strongest and most consistent effects were for happiness. Our findings, therefore, contradict two previous studies, which found that high state anxiety (threat of electric shock) increased recognition accuracy for fearful faces [30,38], compared with low state anxiety. These differences may be due to the different methods of inducing state anxiety. Our findings also challenge other studies that suggest that anxiety is associated with enhanced processing of negative emotions [15,16].

We found no clear evidence of a trait anxiety × emotion interaction on the ability to recognize facial expressions of emotion. This supports previous evidence from our group, which found no clear evidence of a difference in recognition accuracy of fearful facial expressions among individuals reporting high versus low trait anxiety [33], but contradicts other studies which suggest that individuals with high trait anxiety show an enhanced ability to recognize fearful facial expressions [19–21]. Furthermore, we found no clear evidence of a trait anxiety × state anxiety × emotion interaction. Therefore, although state anxiety affected recognition accuracy of specific emotions, trait anxiety did not moderate these effects.

Secondary analyses of sensitivity data showed that high state anxiety led to lower recognition sensitivity to happiness, compared with low state anxiety. We found no clear evidence of an effect of state anxiety on recognition sensitivity for any other emotion. These findings contradict the previous study which found evidence of lower recognition sensitivity in the high state anxiety (versus low state anxiety) condition, for all emotions except happiness (Study 1) and all emotions (Study 2) [39].

The discrepant sensitivity results may be due to the signal detection theory measure of sensitivity used in the current study (A′) compared with unbiased hit rate in the previous study. It could be argued that unbiased hit rate instead measures performance by combining sensitivity and criterion judgements into a single metric. Finally, there was no clear evidence of a relationship between trait anxiety and emotion recognition sensitivity, or an interaction between trait and state anxiety.

This is the first study to investigate the effects of state anxiety on facial emotion processing, and the moderating role of trait anxiety, using the 7.5% $CO_2$ challenge. By experimentally manipulating state anxiety in a controlled laboratory setting, and using a within-subjects design, this study has advantages over previous observational studies that measure between-participant variation in self-reported anxiety. We can better infer the direction of causation between state anxiety and facial emotion processing, and our study does not suffer from confounding due to individual differences. We have also added to previous stress-inducing experimental studies, by using an alternative method of manipulating state anxiety.

The 7.5% $CO_2$ inhalation induces physiological and psychological symptoms akin to generalized anxiety disorder [45], increasing self-reported state anxiety, HR, BP and hypervigilance to threat [45,–47]. The $CO_2$ inhalation may have an impact on cognitive processing related to threat. This is supported by Garner *et al.* [47], who found that it modulates attention mechanisms (i.e. alerting and orienting) involved in the temporal detection and spatial location of salient stimuli. From an experimental perspective, there are benefits of using the 7.5% $CO_2$ inhalation model. First, it yields a reliable unconditioned anxiety response that is less susceptible to individual variation (e.g. compared with models that incite conditioned anxiety responses). Second, unlike some models that induce anxiety and subsequently measure the outcome of interest, the tasks are completed during peak anxiety induction. However, there are different types of anxiety manipulation, and it is possible that experimental tasks that involve a social component (e.g. public speaking) may have different cognitive effects compared with the 7.5% $CO_2$ inhalation, which does not.

Across our three measures of facial emotion processing (emotion recognition accuracy, sensitivity and interpretation bias towards perceiving anger), a pattern emerged for happy facial expressions. Consistent with Attwood *et al.* [39], state anxiety led to lower emotion recognition accuracy particularly for happy facial expressions. Furthermore, in the current study, participants were less sensitive to happy facial expressions, and participants with high trait anxiety showed a greater bias away from perceiving happiness, during heightened state anxiety. Other studies have also found deficits in the processing of happiness among individuals experiencing anxiety. For example, high (versus low) social anxiety predicts slower reaction times for recognizing happiness expressions [26,27] and poorer recognition for faces with happy expressions [68]. However, we acknowledge that accuracy, bias and sensitivity are complex constructs in the field of facial emotion processing, and they can be measured in several different ways.

Our findings may have social and clinical implications. Several social situations that involve facial emotion processing can arouse state anxiety for many people, such as public speaking and interviews. Misinterpreting social cues that are positively valenced during these anxious states could have negative emotional and social consequences. For example, perceiving anger instead of happiness (e.g. by thinking someone is frowning or staring), or failing to detect happiness in neutral or ambiguous faces, could signal rejection or disinterest and lead to negative feelings and emotions in the observer. This could lead to inappropriate or blunted reactions during social interactions or behavioural avoidance, which may evoke negative reactions from others [13], thus potentially impacting attachments and relationships. These findings may be particularly pertinent for people with social anxiety, who experience state anxious episodes across numerous everyday social situations (e.g. talking to people at work or school and meeting strangers), triggered by perceived or actual scrutiny from others [69]. Indeed, some cognitive theories suggest that biases in emotion processing may in turn elicit autonomic arousal and sustain anxious states [70], and biased interpretation of ambiguous social cues is considered a maintenance factor for social anxiety disorder [41]. This vicious cycle may also apply to state anxiety; the impairing effects of state anxiety on facial emotion processing could in turn lead to further anxiety, although we did not test this reverse direction of causality in the current study.

Our findings may also have implications for theory and research. First, the consistency across our measures (accuracy, sensitivity and bias) for happiness but not anger suggests that alterations in the processing of positive (versus negative) emotions may play a more important role in the cognitive aspects of anxiety, a view echoed by other researchers [26]. However, a more solid evidence base is needed to support subsequent theory development. These findings are also interesting given that happy facial expressions are reported to be more easily identified than negative facial expressions [71]. However, we had no *a priori* hypotheses for specific emotions; therefore, strong conclusions cannot be

drawn from our data. The effects of anxiety on the processing of happy facial expressions, and the mechanisms behind a possible impairment, should be specifically investigated in future studies. For example, deficits could be related to the fact that happiness was the only positive signal in the stimulus set and induction of negative affect could lead people to attend to negative information. Second, temperamental differences in anxiety were found to be less impairing than situational fluctuations in anxiety, which highlights the importance of distinguishing between trait and state anxiety in emotion research. This is also encouraging from a clinical perspective, as anxious states are transient and modifiable, compared with trait anxiety which is more immutable. Interventions that help to reduce anxious states could, therefore, also improve facial emotion processing. Although caution should be exercised when extrapolating from one study, and future studies are needed to determine whether some interaction effects (with small sample sizes) are replicable.

Our study has limitations. First, the artificial manipulation of state anxiety and measures of facial emotion processing using computer tasks with static images in a laboratory may reduce the study's ecological validity, and consequently findings may not generalize to other settings. Although the 7.5% $CO_2$ challenge is a well-validated human experimental model of anxiety [45,46], it may not be directly related to social threat situations, where facial emotion processing is most relevant. It would be useful for future studies to examine these questions using state social anxiety manipulations (e.g. via the Trier Social Stress Test). Furthermore, biases in facial emotion processing are context and anxiety specific. Facial cues are clearly less relevant in non-social threat situations and certain anxiety disorders (e.g. specific phobias related to animals or environments). Therefore, there are limits to the generalizability (external validity) of these findings. However, it was necessary for us to use the same experimental manipulation and tasks as the previous study to perform the replication. Otherwise, any discrepant findings could be attributed to methodological differences. Second, the 2AFC task can be interpreted in two ways (i.e. a bias towards anger or a bias away from happiness). Since we did not use additional morph sequences (e.g. happiness–disgust; anger–disgust), we cannot be sure which emotion was driving these effects. However, given the consistency with the 6AFC task results for emotion recognition accuracy, we suggest that the 2AFC task outcomes may have been driven by biases in happiness processing. Due to the limited time available during the inhalation procedure, we could not include several 2AFC tasks. It would be useful for a future study to investigate the effects of state anxiety on interpretation biases to other emotions, to determine whether the results for happiness are unique. Third, we found no clear evidence of a relationship between trait anxiety and emotion processing, which may have been due to the measure used. We recruited participants using an 'extreme groups approach' (low and high trait anxiety scores with the middle tertile excluded), to produce a cleaner, dichotomous measure. However, some researchers are critical of this approach as it may inflate effect sizes and instead argue that continuous measure may increase power and reliability [72–74]. This should be considered in any future replications. Finally, future studies should also examine the role of state and trait anxiety in the speed of facial emotion processing, which may help to elucidate these findings. For example, there is evidence that socially anxious individuals are faster at detecting facial expressions of emotion at moderate (versus low and high) levels of anxious arousal [40].

## 5. Conclusion

Replicating previous research [39], state anxiety reduced emotion recognition accuracy globally, and specifically for happy, disgusted and fearful facial expressions, which suggests these findings are reliable. State anxiety also impaired sensitivity to happy facial expressions, extending previous findings, but this warrants further investigation. We failed to replicate the main effect of state anxiety on interpretation bias towards perceiving anger. In addition, we did not find evidence to indicate a relationship between trait anxiety and emotion recognition accuracy or interpretation bias towards perceiving anger. However, we found some evidence of an interaction; individuals with high trait anxiety displayed an increased tendency to perceive anger and a decreased tendency to perceive happiness when experiencing high but not low state anxiety. Therefore, state and trait anxiety appear to have different effects on facial emotion processing, and it may be necessary to distinguish them in emotion research. Our findings have important psychological implications. When experiencing heightened state anxiety, people exhibit global facial emotion processing deficits, which may have negative emotional and behavioural consequences.

Ethics. Ethical approval was obtained from the Faculty of Science Human Research Ethics Committee at the University of Bristol (reference: 30041520644). All participants provided written informed consent.

Data accessibility. Study data and analysis code are available from the University of Bristol's Research Data Repository (https://data.bris.ac.uk/data/dataset/31ddg7ihazewo2iqggjowqlt7g).

Authors' contributions. A.S.A., I.S.P.-V. and M.R.M. designed the study and wrote the study protocol. A.S.A. oversaw data collection. M.L.D. analysed and interpreted the data and wrote the manuscript. All authors critically revised the manuscript, approved the final version and are accountable for the work.

All authors gave final approval for publication and agreed to be held accountable for the work performed therein.

Competing interests. M.R.M. and I.S.P.-V. are co-directors of Jericoe Ltd, which produces software for the assessment and modification of emotion recognition. We have no other potential competing interests.

Funding. This work was supported by the Medical Research Council Integrative Epidemiology Unit (MRC IEU) at the University of Bristol (grant no. MC_UU_12013/6).

Acknowledgements. We are grateful to the people who participated in this study.

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
