## [Peer Review File · Royal Society Open Science]

Review History

RSOS-210056.R0 (Original submission)

Review form: Reviewer 1

Is the manuscript scientifically sound in its present form?

Yes

Are the interpretations and conclusions justified by the results?

No

Is the language acceptable?

Yes

Do you have any ethical concerns with this paper?

No

Have you any concerns about statistical analyses in this paper?

No

Recommendation?

Major revision is needed (please make suggestions in comments)

Comments to the Author(s)

Review for Royal Society Open Science

Manuscript: RSOS-210056

Title: The Role of State and Trait Anxiety in the Processing of Facial Expressions of Emotion

In this manuscript, the authors report on a study investigating the notion that processing of facial expressions can be hampered by an interplay of trait and state anxiety. Healthy adults high or low in trait anxiety completed tasks measuring emotion recognition accuracy and facial interpretation bias. In a within-subject design the participants completed these tasks once under the influence of a carbon dioxide challenge as state anxiety induction and once while breathing 'hospital oxygen'. They expected (a) that increased state anxiety would decrease emotion recognition accuracy and increase interpretation bias for anger, (b) that increased trait anxiety would also decrease emotion recognition accuracy and increase interpretation bias for anger, and (c) high state and high trait anxiety would increase these effects.

The results indicated that participants with increased state anxiety showed decreased emotion recognition accuracy while no such effect was found on interpretation bias. Elevated levels of trait anxiety by themselves had no impact on emotion recognition nor interpretation bias. When looking at the interaction between state- and trait-anxiety, no significant effects occurred with regard to emotion recognition. Interpretation bias towards anger, however, was increased when state- AND trait-anxiety were high. It also appeared that recognition rates in the high state condition were lower for disgust, fear and particularly happiness.

Additional, explorative analyses with regard to emotion recognition sensitivity revealed a reduced recognition sensitivity for happy faces, while no other effect was significant.

By itself, the subject of the manuscript by itself I find very interesting, if, however, for the readers of RSOS, is the editors decision. The research as such is thoroughly conducted: The screening procedure, the experimental set-up and the analytic approach seem to be pursued with scientific rigor and up to ethical standards. However, to become publishable, the article would need a substantial revision with regard to theoretical imbedding in intro and discussion, textual/language clarity and structure. Please, find more details below.

Abstract:

- Even with a rigorous experimental set-up, I would be always cautious with regard to causal inferences.
- it is not clear that the challenge refers to an anxiety induction
- It is always helpful to report what kind of sample was recruited. Healthy sample recruited from general population ? Male:female ratio ? Students? High/low trait?
- As far as I know it is unusual to report the statistics in the abstract.

Introduction:

The introduction of the relevant theoretical framework is too shallow if not absent. The intro basically consists of short definitions of trait vs state anxiety and a quite thorough collection of studies pleading for or against emotion processing biases when anxiety is concerned. Yet, the whole framework/theoretical bases of processing biases in anxiety is neglected except for one mentioning on page 4, line 43. I think that the general idea of cognitive but at least that of face recognition biases and interpretation biases in particular should be explained in much more depth. What is important for example, is the fact that these biases make sense in specific

situations, but are too prominent in anxiety disorders. What is also very important, is the fact that these biases are context and anxiety specific. For a spider phobic, a faster/more accurate recognition of facial expression makes no sense when in a situation of immanent 'spider-threat', nor does the negative interpretation of a happy face. For a person with social anxiety, faces are the cue to potential rejection and quick recognition or detection of such threat make a lot of sense. Here, misinterpretations of ambiguous faces may even increase state anxiety. The introduction pretty much relies on social anxiety literature, not questioning the relevance of face cues in un-social threat scenarios. Irrespective of the theoretical framework that should justify the choice for these particular processing aspects and the stimulus selection, the choice of state anxiety induction is also noteworthy. I have no doubts that the CO₂ challenge evokes symptoms and distress associated with anxiety states. Yet, it is also a threat clearly based on aversive internal physiological symptoms clearly associated to panic disorder. Why should such an internal state have any effect on improved or biased threat-detection in the environment and particularly for faces when anxiety in general is at stake?

There is also some confusion when recognition accuracy and sensitivity are introduced. The parallels/differences are not clear.

Considering the whole field of face processing (biases) it is also unclear why particularly the interpretation bias is chosen and why particularly that of anger vs happy. In the light of the general claims that are made concerning the influence of anxiety on face processing, this would make more sense when looking at social anxiety, but not necessarily for anxiety in general.

Wouldn't it have made more sense to investigate the general and emotion specific recognition differences per condition and contrast them with the sensitivity measures? Or if the interpretation biases can be theoretically linked contrast recognition and interpretation only and investigate whether recognition and biases (for specific emotions) are related?

With regard to the hypotheses, I would suggest sorting them with regard to the process rather than anxiety type (state vs trait). That would also reflect the structure in the results section. In the light of the contradictory findings reviewed in the intro, the specific hypotheses are not intuitive. In addition, I would also mention the explorative part even without concrete hypothesis.

Methods:

The method seems thorough and accurate but the terminology is confusing. The authors stick to the not quite intuitive acronyms of the tasks they used instead of the concepts they measure. The same counts for the terminology with regard to the anxiety explanation/induction: After a first introduction, I think something like (induced) state anxiety vs control or high vs low state anxiety would make the text much more readable.

I wonder if there was a particular reason to not include 'neutral' in the basic set and as to-be-recognized- expression. Of course this is not an emotion but it would (a) allow to contrast emotions and non-emotion recognition directly, (b) dilute the set of primarily negative emotions and (c) may allow to identify recognition/interpretation biases in one go, e.g., by identifying what people see if the 5% emotional signal is present and neutral is an option.

I'm curious if response times are assessed and if the researchers have looked at speed-accuracy trade-offs. Maybe this trade-off maybe something that changes under stress/anxiety.

I still find the choice for Happy-angry interpretation bias somewhat arbitrary and not convincingly theoretically founded. More combinations or morphs of all emotions with neutral may have shed a clearer picture.

With regard to prescreening and recruitment I wonder whether this study is based on an own sample or if it is part of a bigger dataset and larger study population. Since procedure and set-up are quite similar to the studies from their own lab the authors seek to replicate and repeatedly cite, this is hard to disentangle. To be clear: It is no problem to seek to publish different subsets of a bigger study, but transparency must be warranted.

What was the exact setup of the VAS scales, what were the questions asked and what were the anchors? The MINI is a semi-structured diagnostic interview. To my knowledge it is not officially translated to a self-report version.

With regard to the inhalations it is not clear whether participants wear the masks/inhale throughout the whole time or only at the beginning of the task and at what point the state measures took place.

Results:

The authors should consider putting the means of the participant characteristics in the same table as the comparisons of the state measures (table 3?) rather than in the text. In addition, they should also statistically compare the general characteristics between groups to verify that the stratification with regard to trait has not brought along any other unwanted group differences except the expected ones.

I would also structure the results in line with the mentioning of the tasks. Up until here the recognition task is always mentioned first and then the interpretation task is mentioned. In the results section it is the other way around. Besides that, as mentioned in the intro, it would make sense to sort the hypothesis by task rather than by anxiety-type. That would also give the manuscript a much clearer structure.

The authors talk about 'some evidence' for 'significant results' and 'no clear evidence' for non-significant findings even when far off non-significant 'trends'. They also tend to interpret the differences in the mean scores in non-significant findings. Despite being incorrect, this framing of the results is also misleading. Please, talk about, eg., significant differences vs no differences, effect vs no effect, or something alike. The description of the high-state anxiety condition as 'gas (i.e., state anxiety)' is confusing. Please, consider comprehensive rephrasing here and throughout the manuscript.

Also should the statistical results be translated to 'understandable' language without interpreting them here in the results: e.g., 'there was some evidence of a difference in threshold scores between low vs high trait anxiety scores'. Does that mean something like: 'there was a tendency in the high trait anxiety group to interpret happy as angry'?

The paragraph about the 6AFC starts with indicating a 2x2 model while the first result presented is a main effect. That is confusing.

The authors should consider (most recent) APA norms for reporting results and take into account when the zero before a decimal point is reported and when not.

On page 14 line 56 they talk about a smaller effect size while indicating earlier that it was strong evidence. I feel that the evaluation of how strong a particular effect is should be done in the discussion section. Here, it should be merely reported.

To me it appears that the evaluation of fewer hits 'particularly' for happiness may result from a comparably subjective evaluation if effect-sizes are not taken into account.

The sensitivity results should be marked more clearly as exploratory here.

The manipulation check is a valuable addition. Here, the participant stratification could be mentioned as well, if not done earlier. It would also be valuable to report the results of 'counterbalancing' analyses here. Did the order of anxiety manipulation and/or tasks make a difference?

Discussion:

In general, I find the discussion (as the introduction) shallow and confusing with regard to straightforward terminology and theoretical purpose. Facial emotion processing in the context of anxiety is used too broad when only few aspects are taken into account.

It is confusing that the authors talk of their first hypothesis being confirmed while later on they say it is not. Maybe they should consider separating the hypotheses per task (as suggested earlier) and structure their results and discussion section accordingly.

On p16 lines 12ff, non-significant differences of mean scores are interpreted.

On p16 lines 33ff, the conclusion is rather far-fetched considering the fact that only one emotion pair was tested. I strongly suggest to strive for more theoretical imbedding of the results.

Explanation of the results are primarily sought in technical and methodological differences rather than in underlying mechanisms and theoretical predictions. It seems at times that primarily earlier work of the authors is taken as reference point rather than any theoretical framework.

The fact, for instance, that no main effect of trait anxiety was detected makes a lot of theoretical sense: It is a latent trait and it has been argued that the associated processing patterns are only activated when the (more frequently) occurring anxiety states are triggered in the high-trait anxious individuals. But again, anxiety inducing suffocation signals may not be directly related to processes going on in fears of negative evaluation, fear of spiders, etc. On the other hand, threatening facial expressions are not necessarily relevant in situations when one fears to suffocate. These aspects should be disentangled.

In addition, the authors could, e.g., discuss if their 'happiness' findings couldn't be related to the fact that happy is actually their only positive signal in the stimulus set. It could appear as the 'odd-one out' or are in general primed with a negative mindset especially in a threatening situation influencing their choices in the tasks.

The methodology is very interesting and surely has potential to be useful for anxiety research, but with a lacking theoretical bases the assumption that the results may be relevant for understanding anxiety disorders is unfortunately far-fetched.

Technical issues:

The paper could be more structured and in more depth.

The style of writing is fine but terminology could be more straight-forward.

Several APA errors with regard to reporting statistics are observed.

Table 2 could be more condensed.

Table 3 could become part of a Table 1 in which the population descriptive means per group are depicted as well as those of the state measures. The results of difference testing could be added there as well.

In sum, I have my doubts that the manuscript should be published in its current state. Theoretical bases and clearer terminology should be provided to justify the choices made for the current set-up. In the end the results should be discussed in the light of the theories again, to show in how far they have increased our insight in the mechanisms of anxiety and potential impact for the clinical field.

Review form: Reviewer 2

Is the manuscript scientifically sound in its present form?

Yes

Are the interpretations and conclusions justified by the results?

Yes

Is the language acceptable?

Yes

Do you have any ethical concerns with this paper?

No

Have you any concerns about statistical analyses in this paper?

No

Recommendation?

Accept with minor revision (please list in comments)

Comments to the Author(s)

Thank you for asking me to review this very interesting manuscript. The paper is about the role of state and trait anxiety on facial emotion processing. The paper is very well written. The background is outlined clearly, the used tasks are explained well, statistical analyses are clear, changes compared to a previously published study protocol are explained, results are reported clearly, and the discussion covers all results. I only have some minor comments:

1. The used questionnaires are mentioned only briefly. I suspect more information can be found in the study protocol. This is fine but it would be transparent to report the psychometric properties, at least the reliability, of the tests used in the current study.
2. In the method section, page 10, line 47, abbreviations BP and HR are used. Even though these abbreviations are well known, they need to be written out/introduced.
3. In the method section, page 12, line 44-47, it is stated that a previously proposed analysis appeared to be insufficient. Therefore, 6 separate 2×2 models were examined. I am wondering whether the authors applied any correction for multiple testing? As there were quite some statistical analyses conducted in a relatively small sample, this should have been done.
4. Throughout the manuscript, the interaction effects are indicated by using "x" instead of a multiplication sign. E.g. gas x trait anxiety instead of gas \times trait anxiety
5. In the discussion, page 16, line 12 - 18, the authors state that there was no clear evidence of a difference in interpretation bias but they interpreted the direction of the results, which was in correspondence with their expectations. However, the effect was statistically not significant. This means that also no trends can be observed and interpreted.
6. No comment just a thought out of curiosity. In the literature about hostility biases, it has been suggested that aggressive individuals experience difficulties in processing social/emotional information because it is inconsistent with their (cognitive) schemas. They need more time to process schema inconsistent information because it differs from their expectations. Furthermore, due to a high emotionality, they may experience any more difficulties to assess the situation from different perspectives. Resulting in relying more on existing schema's. In turn, this makes to interpretation of social information in a hostile manner more likely. I was wondering, whether such a mechanism could also be present in the case of anxiety? It would be interesting to discover whether such (or other) underlying mechanisms apply to biases in social/emotional information processing across psychopathologies.

Decision letter (RSOS-210056.R0)

Dear Dr Dyer

The Editors assigned to your paper RSOS-210056 "The Role of State and Trait Anxiety in the Processing of Facial Expressions of Emotion" have now received comments from reviewers and would like you to revise the paper in accordance with the reviewer comments and any comments from the Editors. Please note this decision does not guarantee eventual acceptance.

Please submit your revised manuscript and required files (see below) no later than 21 days from today's (ie 21-May-2021) date. Note: the ScholarOne system will 'lock' if submission of the revision is attempted 21 or more days after the deadline. If you do not think you will be able to meet this deadline please contact the editorial office immediately.

on behalf of Dr Inti Brazil (Associate Editor) and Essi Viding (Subject Editor)
openscience@royalsociety.org

Reviewer comments to Author:
Reviewer: 1
Comments to the Author(s)
Review for Royal Society Open Science

Manuscript: RSOS-210056
Title: The Role of State and Trait Anxiety in the Processing of Facial Expressions of Emotion

In this manuscript, the authors report on a study investigating the notion that processing of facial expressions can be hampered by an interplay of trait and state anxiety. Healthy adults high or low in trait anxiety completed tasks measuring emotion recognition accuracy and facial interpretation bias. In a within-subject design the participants completed these tasks once under the influence of a carbon dioxide challenge as state anxiety induction and once while breathing 'hospital oxygen'. They expected (a) that increased state anxiety would decrease emotion

recognition accuracy and increase interpretation bias for anger, (b) that increased trait anxiety would also decrease emotion recognition accuracy and increase interpretation bias for anger, and (c) high state and high trait anxiety would increase these effects.

The results indicated that participants with increased state anxiety showed decreased emotion recognition accuracy while no such effect was found on interpretation bias. Elevated levels of trait anxiety by themselves had no impact on emotion recognition nor interpretation bias. When looking at the interaction between state- and trait-anxiety, no significant effects occurred with regard to emotion recognition. Interpretation bias towards anger, however, was increased when state- AND trait-anxiety were high. It also appeared that recognition rates in the high state condition were lower for disgust, fear and particularly happiness.

Additional, explorative analyses with regard to emotion recognition sensitivity revealed a reduced recognition sensitivity for happy faces, while no other effect was significant.

By itself, the subject of the manuscript by itself I find very interesting, if, however, for the readers of RSOS, is the editors decision. The research as such is thoroughly conducted: The screening procedure, the experimental set-up and the analytic approach seem to be pursued with scientific rigor and up to ethical standards. However, to become publishable, the article would need a substantial revision with regard to theoretical imbedding in intro and discussion, textual/language clarity and structure. Please, find more details below.

Abstract:

- Even with a rigorous experimental set-up, I would be always cautious with regard to causal inferences.
- it is not clear that the challenge refers to an anxiety induction
- It is always helpful to report what kind of sample was recruited. Healthy sample recruited from general population ? Male:female ratio ? Students? High/low trait?
- As far as I know it is unusual to report the statistics in the abstract.

Introduction:

The introduction of the relevant theoretical framework is too shallow if not absent. The intro basically consists of short definitions of trait vs state anxiety and a quite thorough collection of studies pleading for or against emotion processing biases when anxiety is concerned. Yet, the whole framework/theoretical bases of processing biases in anxiety is neglected except for one mentioning on page 4, line 43. I think that the general idea of cognitive but at least that of face recognition biases and interpretation biases in particular should be explained in much more depth. What is important for example, is the fact that these biases make sense in specific situations, but are too prominent in anxiety disorders. What is also very important, is the fact that these biases are context and anxiety specific. For a spider phobic, a faster/more accurate recognition of facial expression makes no sense when in a situation of immanent 'spider-threat', nor does the negative interpretation of a happy face. For a person with social anxiety, faces are the cue to potential rejection and quick recognition or detection of such threat make a lot of sense. Here, misinterpretations of ambiguous faces may even increase state anxiety. The introduction pretty much relies on social anxiety literature, not questioning the relevance of face cues in un-social threat scenarios. Irrespective of the theoretical framework that should justify the choice for these particular processing aspects and the stimulus selection, the choice of state anxiety induction is also noteworthy. I have no doubts that the CO2 challenge evokes symptoms and distress associated with anxiety states. Yet, it is also a threat clearly based on aversive internal physiological symptoms clearly associated to panic disorder. Why should such an internal state have any effect on improved or biased threat-detection in the environment and particularly for faces when anxiety in general is at stake?

There is also some confusion when recognition accuracy and sensitivity are introduced. The parallels/ differences are not clear.

Considering the whole field of face processing (biases) it is also unclear why particularly the interpretation bias is chosen and why particularly that of anger vs happy. In the light of the general claims that are made concerning the influence of anxiety on face processing, this would make more sense when looking at social anxiety, but not necessarily for anxiety in general.

Wouldn't it have made more sense to investigate the general and emotion specific recognition differences per condition and contrast them with the sensitivity measures? Or if the interpretation biases can be theoretically linked contrast recognition and interpretation only and investigate whether recognition and biases (for specific emotions) are related?

With regard to the hypotheses, I would suggest sorting them with regard to the process rather than anxiety type (state vs trait). That would also reflect the structure in the results section. In the light of the contradictory findings reviewed in the intro, the specific hypotheses are not intuitive. In addition, I would also mention the explorative part even without concrete hypothesis.

Methods:

The method seems thorough and accurate but the terminology is confusing. The authors stick to the not quite intuitive acronyms of the tasks they used instead of the concepts they measure. The same counts for the terminology with regard to the anxiety explanation/induction: After a first introduction, I think something like (induced) state anxiety vs control or high vs low state anxiety would make the text much more readable.

I wonder if there was a particular reason to not include 'neutral' in the basic set and as to-be-recognized-expression. Of course this is not an emotion but it would (a) allow to contrast emotions and non-emotion recognition directly, (b) dilute the set of primarily negative emotions and (c) may allow to identify recognition/interpretation biases in one go, e.g., by identifying what people see if the 5% emotional signal is present and neutral is an option.

I'm curious if response times are assessed and if the researchers have looked at speed-accuracy trade-offs. Maybe this trade-off maybe something that changes under stress/anxiety.

I still find the choice for Happy-angry interpretation bias somewhat arbitrary and not convincingly theoretically founded. More combinations or morphs of all emotions with neutral may have shed a clearer picture.

With regard to prescreening and recruitment I wonder whether this study is based on an own sample or if it is part of a bigger dataset and larger study population. Since procedure and set-up are quite similar to the studies from their own lab the authors seek to replicate and repeatedly cite, this is hard to disentangle. To be clear: It is no problem to seek to publish different subsets of a bigger study, but transparency must be warranted.

What was the exact setup of the VAS scales, what were the questions asked and what were the anchors? The MINI is a semi-structured diagnostic interview. To my knowledge it is not officially translated to a self-report version.

With regard to the inhalations it is not clear whether participants wear the masks/inhale throughout the whole time or only at the beginning of the task and at what point the state measures took place.

Results:

The authors should consider putting the means of the participant characteristics in the same table as the comparisons of the state measures (table 3?) rather than in the text. In addition, they should also statistically compare the general characteristics between groups to verify that the stratification with regard to trait has not brought along any other unwanted group differences except the expected ones.

I would also structure the results in line with the mentioning of the tasks. Up until here the recognition task is always mentioned first and then the interpretation task is mentioned. In the results section it is the other way around. Besides that, as mentioned in the intro, it would make sense to sort the hypothesis by task rather than by anxiety-type. That would also give the manuscript a much clearer structure.

The authors talk about 'some evidence' for 'significant results and 'no clear evidence' for non-significant findings even when far off non-significant 'trends'. They also tend to interpret the differences in the mean scores in non-significant findings. Despite being incorrect, this framing of the results is also misleading. Please, talk about, eg., significant differences vs no differences, effect vs no effect, or something alike. The description of the high-state anxiety condition as 'gas (i.e., state anxiety)' is confusing. Please, consider comprehensive rephrasing here and throughout the manuscript.

Also should the statistical results be translated to 'understandable' language without interpreting them here in the results: e.g., 'there was some evidence of a difference in threshold scores between low vs high trait anxiety scores'. Does that mean something like: 'there was a tendency in the high trait anxiety group to interpret happy as angry'?

The paragraph about the 6AFC starts with indicating a 2x2 model while the first result presented is a main effect. That is confusing.

The authors should consider (most recent) APA norms for reporting results and take into account when the zero before a decimal point is reported and when not.

On page 14 line 56 they talk about a smaller effect size while indicating earlier that it was strong evidence. I feel that the evaluation of how strong a particular effect is should be done in the discussion section. Here, it should be merely reported.

To me it appears that the evaluation of fewer hits 'particularly' for happiness may result from a comparably subjective evaluation if effect-sizes are not taken into account.

The sensitivity results should be marked more clearly as exploratory here.

The manipulation check is a valuable addition. Here, the participant stratification could be mentioned as well, if not done earlier. It would also be valuable to report the results of 'counterbalancing' analyses here. Did the order of anxiety manipulation and/or tasks make a difference?

Discussion:

In general, I find the discussion (as the introduction) shallow and confusing with regard to straightforward terminology and theoretical purpose. Facial emotion processing in the context of anxiety is used too broad when only few aspects are taken into account.

It is confusing that the authors talk of their first hypothesis being confirmed while later on they say it is not. Maybe they should consider separating the hypotheses per task (as suggested earlier) and structure their results and discussion section accordingly.

On p16 lines 12ff, non-significant differences of mean scores are interpreted.

On p16 lines 33ff, the conclusion is rather far-fetched considering the fact that only one emotion pair was tested. I strongly suggest to strive for more theoretical imbedding of the results.

Explanation of the results are primarily sought in technical and methodological differences rather than in underlying mechanisms and theoretical predictions. It seems at times that primarily earlier work of the authors is taken as reference point rather than any theoretical framework.

The fact, for instance, that no main effect of trait anxiety was detected makes a lot of theoretical sense: It is a latent trait and it has been argued that the associated processing patterns are only activated when the (more frequently) occurring anxiety states are triggered in the high-trait anxious individuals. But again, anxiety inducing suffocation signals may not be directly related to processes going on in fears of negative evaluation, fear of spiders, etc. On the other hand,

threatening facial expressions are not necessarily relevant in situations when one fears to suffocate. These aspects should be disentangled.

In addition, the authors could, e.g., discuss if their 'happiness' findings couldn't be related to the fact that happy is actually their only positive signal in the stimulus set. It could appear as the 'odd-one out' or are in general primed with a negative mindset especially in a threatening situation influencing their choices in the tasks.

The methodology is very interesting and surely has potential to be useful for anxiety research, but with a lacking theoretical bases the assumption that the results may be relevant for understanding anxiety disorders is unfortunately far-fetched.

Technical issues:

The paper could be more structured and in more depth.

The style of writing is fine but terminology could be more straight-forward.

Several APA errors with regard to reporting statistics are observed.

Table 2 could be more condensed.

Table 3 could become part of a Table 1 in which the population descriptive means per group are depicted as well as those of the state measures. The results of difference testing could be added there as well.

In sum, I have my doubts that the manuscript should be published in its current state. Theoretical bases and clearer terminology should be provided to justify the choices made for the current set-up. In the end the results should be discussed in the light of the theories again, to show in how far they have increased our insight in the mechanisms of anxiety and potential impact for the clinical field.

Reviewer: 2

Comments to the Author(s)

Thank you for asking me to review this very interesting manuscript. The paper is about the role of state and trait anxiety on facial emotion processing. The paper is very well written. The background is outlined clearly, the used tasks are explained well, statistical analyses are clear, changes compared to a previously published study protocol are explained, results are reported clearly, and the discussion covers all results. I only have some minor comments:

1. The used questionnaires are mentioned only briefly. I suspect more information can be found in the study protocol. This is fine but it would be transparent to report the psychometric properties, at least the reliability, of the tests used in the current study.
2. In the method section, page 10, line 47, abbreviations BP and HR are used. Even though these abbreviations are well known, they need to be written out/introduced.
3. In the method section, page 12, line 44-47, it is stated that a previously proposed analysis appeared to be insufficient. Therefore, 6 separate 2×2 models were examined. I am wondering whether the authors applied any correction for multiple testing? As there were quite some statistical analyses conducted in a relatively small sample, this should have been done.
4. Throughout the manuscript, the interaction effects are indicated by using "x" instead of a multiplication sign. E.g. gas x trait anxiety instead of gas × trait anxiety
5. In the discussion, page 16, line 12 - 18, the authors state that there was no clear evidence of a difference in interpretation bias but they interpreted the direction of the results, which was in correspondence with their expectations. However, the effect was statistically not significant. This means that also no trends can be observed and interpreted.
6. No comment just a thought out of curiosity. In the literature about hostility biases, it has been suggested that aggressive individuals experience difficulties in processing social/emotional information because it is inconsistent with their (cognitive) schemas. They need more time to

process schema inconsistent information because it differs from their expectations. Furthermore, due to a high emotionality, they may experience any more difficulties to assess the situation from different perspectives. Resulting in relying more on existing schema's. In turn, this makes to interpretation of social information in a hostile manner more likely. I was wondering, whether such a mechanism could also be present in the case of anxiety? It would be interesting to discover whether such (or other) underlying mechanisms apply to biases in social/emotional information processing across psychopathologies.

===PREPARING YOUR MANUSCRIPT===

===PREPARING YOUR REVISION IN SCHOLARONE===

Please ensure that you include a summary of your paper at Step 2 'Type, Title, & Abstract'. This should be no more than 100 words to explain to a non-scientific audience the key findings of your

research. This will be included in a weekly highlights email circulated by the Royal Society press office to national UK, international, and scientific news outlets to promote your work.

Author's Response to Decision Letter for (RSOS-210056.R0)

See Appendix A.

RSOS-210056.R1 (Revision)

Review form: Reviewer 1

Is the manuscript scientifically sound in its present form?

Yes

Are the interpretations and conclusions justified by the results?

Yes

Is the language acceptable?

Yes

Do you have any ethical concerns with this paper?

No

Have you any concerns about statistical analyses in this paper?

No

Recommendation?

Major revision is needed (please make suggestions in comments)

Comments to the Author(s)

Review for Royal Society Open Science

Manuscript: RSOS-210056.R1

Title: The Role of State and Trait Anxiety in the Processing of Facial Expressions of Emotion

By revising their manuscript the authors have improved their manuscript considerably with regard to structure and readability. Yet, I am still not convinced about the theoretical embedding of the study. In particular the authors do introduce two models in the introduction (Gray as well as Williams et al.) Both models suggest that state as well as trait anxiety should improve or prioritize the processing of fear relevant cues. The fact that certain traits lead to overexaggerated (rapid detection of supposed threat, or seeing threat where is none, difficulty disengaging) nicely fit in that idea. Even quick recognition of threat makes sense here, as it may be that hyper alertness increases perceptual vigilance. You could say that anxiety pathology seems to be an over-sensitivity of potential threat detection and its processing. While quick processing of real threat seems to have a possible evolutionary advantage, it may be disadvantageous when specific anxious traits may bias an organism in preferentially 'looking' for the feared object in question (leading to chronic anxiety related stress and increases in biased processing). Thus from an evolutionary AND from a psychopathological standpoint preferential processing of (potentially) threatening stimuli make sense. In my eyes, the contradictory (previous) findings (of the authors) need to be seen in the light of these models rather than merely summing them up. The mentioned meta-analysis of Demenescu, for instance DOES indicate a moderate deficit of face recognition in anxiety disorders, BUT they do discuss these findings in the light of the models in the sense that advantages of threat processing in one domain (eg. vigilance/quick detection) may come at the cost of deficits in another domain (eg. overgeneralization/interpretation of negative as threat or erroneous categorization/recognition). To my knowledge all articles that present conflicting results do at least some attempts to explain them in the light of the current/ dominant theories or try to nuance them by suggesting alternative theoretical explanations. The current paper does not do that. In addition, the suggestion that erroneous face processing may lead to problems in social interaction is, apart from socially anxious individuals, hardly the problem. The current study

should primarily increase mechanistic or maybe methodological understanding of threat processing in anxiety. [E.g., is it possible that the method used is very reliable in inducing state anxiety or 'symptoms' related to it but that the potential changes in CO₂ may have a physiological impact on cognitive processing irrigated to fear? Could it be possible that lab induced physiological fear states that are with no respect related to the task at hand are fundamentally different from fear states used in experimental work such as 'announcements of a speech to be held after the task (for socially anxious individuals)' even though physiological measures may be identical? What would that mean for our theoretical and mechanistically understanding of state anxiety across the different anxiety disorders and in general?]

I am convinced that the authors have a solid knowledge of experimental and statistical techniques striving for useful and necessary replication of their earlier work but I cannot help the notion that they are either not aware of or are somewhat negligent with thoroughly seeking out the theoretical basis for their project, their hypotheses and implications of their results. As I read the introduction the authors are not very precise in defining and distinguishing the biases they want to investigate particularly with regard to facial expressions. Recognition, sensitivity and interpretation are neither discussed separately nor is their possible interconnectedness mentioned. The terminology throughout the manuscript gives more evidence of this omission: what is an increased bias for anger - is an angry expression seen as MORE angry than it actually is? How can a bias of happy be decreased? At what point is the absence of any bias established in this work? In the design section the authors speak of 'emotional bias' and later (in the discussion) of an anger emotion processing bias and an anger bias. Also in the discussion, they talk of three measures of facial emotion processing: accuracy, sensitivity and bias, again showing quite some indifference with regard to the nuances. In the field of (biased) face processing, accuracy and sensitivity can also relate to, eg. probe detection in relation to threat faces detection or location of specific expressions in a grid of faces. Bias can relate to literally any aspect of face processing.

In sum, I would strongly advice the authors to discuss their work with a colleague who is highly familiar with cognitive biases in anxiety and particularly face processing biases.

Minor points:

- The hypotheses are stated in a much clearer way in the preregistration than they are in this manuscript. A point by point write-up may increase clarity
- The use of the cryptic task Acronyms has been changed to some degree but not consistently. They seem to come back in tables, figures and throughout the manuscript. The same counts for the 7.5%CO₂ vs Air descriptors.
- Some hints on literature that may help the authors in understanding the framework of cognitive processing in anxiety and potential use of trait vs state difference. There are more and also more recent ones, but that should give the authors an idea of what to look for ...

MacLeod, C., & Rutherford, E. (1992). Anxiety and the selective processing of emotional information: Mediating roles of awareness, trait and state variables, and personal relevance of stimulus materials. *Behaviour Research and Therapy*, 30(5), 479-491.

Schulz, S. M., Alpers, G. W., & Hofmann, S. G. (2008). Negative self-focused cognitions mediate the effect of trait social anxiety on state anxiety [Article]. *Behaviour Research and Therapy*, 46(4), 438-449. <Go to ISI>://000255316200003

Helzer, E. G., Connor-Smith, J. K., & Reed, M. A. (2009). Traits, states, and attentional gates: Temperament and threat relevance as predictors of attentional bias to social threat. *Anxiety, Stress & Coping*, 22(1), 57 - 76. <http://www.informaworld.com/10.1080/10615800802272244>

Review form: Reviewer 3

Is the manuscript scientifically sound in its present form?

Yes

Are the interpretations and conclusions justified by the results?

Yes

Is the language acceptable?

Yes

Do you have any ethical concerns with this paper?

No

Have you any concerns about statistical analyses in this paper?

No

Recommendation?

Accept with minor revision (please list in comments)

Comments to the Author(s)

Dear authors,

Thank you for your work on how trait and state anxiety modulate emotional face processing. Considering that the literature is muddled in contradictory results, more robust, clear, and pre-registered studies such as this one are much needed. As other reviewers have already provided much feedback, I just have a few minor comments:

- Abstract: the authors should report all results for state anxiety and then for trait anxiety, as this would make for more logical description of the results. This would result in the following structure:

“The strongest emotion-specific effects were for happiness, with reduced accuracy ($p = .002$, $d_z = 0.49$) and sensitivity ($p = .004$, $\eta^2 = .17$) during heightened state anxiety. However, there was evidence of increased anger bias and decreased happiness bias during heightened state anxiety, among individuals with high trait anxiety ($p = .03$).”

Furthermore, the “biases” should be phrased otherwise or explained within parentheses at some point of the abstract to clarify what is meant, e.g., “[state anxiety] appears to facilitate the processing of anger but impair the detection of happy expressions.” or something along those lines.

- The definition of trait and state anxiety deserves a sentence of its own, so this should be separated from the rest of the sentence: “are transient reactions to environmental stressors (31, 32).”

- Power calculations: the authors should report the power afforded by the current sample assuming a d of .69.

- Internal consistency estimates (preferably McDonald’s Omega) should be reported for all measures of interest, including both questionnaire and task-based measures.

- All p-values should be reported to the third decimal place, following APA good practice recommendations.

- I am missing a discussion on the potential behavioral consequences of the observed biases. Without straying too far from the data, the authors should briefly discuss how the observed biases might play out in terms of real-life approach-avoidance behavior. This would also dovetail with the Introduction, where the authors do comment on these issues. Moreover, the authors should more precisely delimit the relevance of the reported effects for anxiety in comparison to other psychopathological dimensions, as similar biases have been reported for individuals high on trait anger and aggression.

Review form: Reviewer 4

Is the manuscript scientifically sound in its present form?

Yes

Are the interpretations and conclusions justified by the results?

Yes

Is the language acceptable?

Yes

Do you have any ethical concerns with this paper?

No

Have you any concerns about statistical analyses in this paper?

No

Recommendation?

Accept with minor revision (please list in comments)

Comments to the Author(s)

The current manuscript compares the interactive influence of state and trait anxiety on facial expression recognition. The experimental manipulation of state anxiety (i.e, 7.5% carbon dioxide challenge) as well as the assessment of emotion recognition (biases) with 6AFC and 2AFC tasks is innovative and clearly improves our understanding of underlying psychological mechanisms. The manuscript in its current version demands a lot of inference from the reader and arguments are often not formulated clearly. I have some minor comments:

Abstract:

- As reviewer 1 already mentioned, it is unusual to report statistics in such an amount in the abstract as it decreases readability. If you really want such information in the abstract, please be consistent and report effect sizes to all p-values and use either two or three decimal places
- Reporting the place of recruitment in the abstract is also highly unusual and doesn't add valuable information in my opinion
- I don't think that "7.5% carbon dioxide challenge" is a widely known paradigm, maybe you can enter a short descriptive sentence about this experimental manipulation including its sham equivalent instead

Introduction:

- Last paragraph p4: "Measurement differences may also explain mixed findings. In general, accuracy reflects hits (correct identification of emotions), bias reflects hits and false alarms (incorrect identification of emotions), and sensitivity reflects hits whilst accounting for false alarms." □ Statistic might be a better word compared to measurement. This whole paragraph is quite confusing and demands a lot of inference by the reader. Please clarify your arguments.

- P5L5: What is "naturally occurring state Anxiety"?

- P5L18: What is "7.5% carbon dioxide (CO₂) challenge"? Maybe you don't have to define this paradigm here, but can describe the conceptual idea "participants inhale XXX to XXX "

Method:

- P7L39: "Participants were eligible if they met our criteria for low or high trait anxiety □ Which are? Please refer to the paragraph where you describe your criteria

- P8L10: "Scores between 21-31 and 44-64 denoted low and high trait anxiety, respectively. For the last two participants, we lowered the threshold of inclusion for high trait anxiety to 41-64 to support recruitment. □ Very complicated, maybe only: "Scores between 21-31 and 41-64 denoted low and high"?

- P9L26: Maybe you like to add the following reference which found that neutral and angry faces elicit comparable negative facial responses when they are passively viewed which indicates the negative valence of neutral faces: Höfling, T., Alpers, G. W., Gerdes, A. B. M., & Föhl, U. (2021). Automatic Facial Coding Versus Electromyography of Mimicked, Passive and Inhibited Facial Response to Emotional Faces. *Cognition and Emotion*, 35(5), 874-889.

<https://doi.org/10.1080/02699931.2021.1902786>

- P10L12: What is "A-prime"?

- ANOVA is a common abbreviation

Results:

- Again, varying decimal place of p-values, please unify

- "There was also strong evidence of a state anxiety emotion interaction [$F(3.76, 165.36) = 4.11, p = .004, \eta^2 p^2 = .09$]. □ This is surely not a large effect, please indicate your effect size interpretation in the corresponding method section

- The assumption of sphericity was violated for the emotion, and state anxiety × emotion within-subjects effects, so here we report Greenhouse Geisser statistics. □ Such information needs to be in the corresponding method section and not in the results section

- P16L33: non-significant differences of mean scores are interpreted. There is clearly no difference between them.

- Please move your manipulation check from the end to the beginning of the results (order of tables is currently wrong too)

- "gas inhalation order modified the effect of state anxiety on emotion recognition accuracy ($p = .01$), with stronger effects when the 7.5% CO₂ inhalation came first." □ Is the robustness of your findings threatened? Please provide statistics and more details on this effect as a supplement

Decision letter (RSOS-210056.R1)

Dear Dr Dyer

The Editors assigned to your paper RSOS-210056.R1 "The Role of State and Trait Anxiety in the Processing of Facial Expressions of Emotion" have now received comments from reviewers and would like you to revise the paper in accordance with the reviewer comments and any comments from the Editors. Please note this decision does not guarantee eventual acceptance.

It is unusual that a further round of revisions have been offered, so we urge you to make every effort to fully address all of the comments at this stage, as further revisions may not be possible. If deemed necessary by the Editors, your manuscript will be sent back to one or more of the original reviewers for assessment. If the original reviewers are not available, we may invite new reviewers.

Please submit your revised manuscript and required files (see below) no later than 21 days from today's (ie 29-Sep-2021) date. Note: the ScholarOne system will 'lock' if submission of the revision is attempted 21 or more days after the deadline. If you do not think you will be able to meet this deadline please contact the editorial office immediately.

on behalf of Dr Inti Brazil (Associate Editor) and Essi Viding (Subject Editor)
openscience@royalsociety.org

Associate Editor Comments to Author (Dr Inti Brazil):

Associate Editor: 1

Comments to the Author:

Dear dr. Dyer,

The reviewers from the previous round were mixed in their re-evaluation of the revised manuscript. I decided to secure additional reviews from other experts. As you will see, multiple reviewers highlight that there are still major conceptual issues and that the theoretical embedding is not sufficiently developed. One of the reviewers found it particularly difficult to follow the line of reasoning, as it requires a lot of inferencing from the reader. There are also methodological

points that require attention. I believe that the comments are clearly articulated and can be used to improve the manuscript.

Reviewer comments to Author:

Reviewer: 1

Comments to the Author(s)

Review for Royal Society Open Science

Manuscript: RSOS-210056.R1

Title: The Role of State and Trait Anxiety in the Processing of Facial Expressions of Emotion

By revising their manuscript the authors have improved their manuscript considerably with regard to structure and readability. Yet, I am still not convinced about the theoretical embedding of the study. In particular the authors do introduce two models in the introduction (Gray as well as Williams et al.) Both models suggest that state as well as trait anxiety should improve or prioritize the processing of fear relevant cues. The fact that certain traits lead to overexaggerated (rapid detection of supposed threat, or seeing threat where is none, difficulty disengaging) nicely fit in that idea. Even quick recognition of threat makes sense here, as it may be that hyper alertness increases perceptual vigilance. You could say that anxiety pathology seems to be an over-sensitivity of potential threat detection and its processing. While quick processing of real threat seems to have a possible evolutionary advantage, it may be disadvantageous when specific anxious traits may bias an organism in preferentially 'looking' for the feared object in question (leading to chronic anxiety related stress and increases in biased processing). Thus from an evolutionary AND from a psychopathological standpoint preferential processing of (potentially) threatening stimuli make sense. In my eyes, the contradictory (previous) findings (of the authors) need to be seen in the light of these models rather than merely summing them up. The mentioned meta-analysis of Demenescu, for instance DOES indicate a moderate deficit of face recognition in anxiety disorders, BUT they do discuss these findings in the light of the models in the sense that advantages of threat processing in one domain (eg. vigilance/quick detection) may come at the cost of deficits in another domain (eg. overgeneralization/interpretation of negative as threat or erroneous categorization/recognition). To my knowledge all articles that present conflicting results do at least some attempts to explain them in the light of the current/dominant theories or try to nuance them by suggesting alternative theoretical explanations. The current paper does not do that. In addition, the suggestion that erroneous face processing may lead to problems in social interaction is, apart from socially anxious individuals, hardly the problem. The current study should primarily increase mechanistic or maybe methodological understanding of threat processing in anxiety. [E.g., is it possible that the method used is very reliable in inducing state anxiety or 'symptoms' related to it but that the potential changes in CO2 may have a physiological impact on cognitive processing irrigated to fear? Could it be possible that lab induced physiological fear states that are with no respect related to the task at hand are fundamentally different from fear states used in experimental work such as 'announcements of a speech to be held after the task (for socially anxious individuals)' even though physiological measures may be identical? What would that mean for our theoretical and mechanistically understanding of state anxiety across the different anxiety disorders and in general?]

I am convinced that the authors have a solid knowledge of experimental and statistical techniques striving for useful and necessary replication of their earlier work but I cannot help the notion that they are either not aware of or are somewhat negligent with thoroughly seeking out the theoretical basis for their project, their hypotheses and implications of their results. As I read the introduction the authors are not very precise in defining and distinguishing the biases they want to investigate particularly with regard to facial expressions. Recognition, sensitivity and interpretation are neither discussed separately nor is their possible interconnectedness

mentioned. The terminology throughout the manuscript gives more evidence of this omission: what is an increased bias for anger - is an angry expression seen as MORE angry than it actually is? How can a bias of happy be decreased? At what point is the absence of any bias established in this work? In the design section the authors speak of 'emotional bias' and later (in the discussion) of an anger emotion processing bias and an anger bias. Also in the discussion, they talk of three measures of facial emotion processing: accuracy, sensitivity and bias, again showing quite some indifference with regard to the nuances. In the field of (biased) face processing, accuracy and sensitivity can also relate to, eg. probe detection in relation to threat faces detection or location of specific expressions in a grid of faces. Bias can relate to literally any aspect of face processing.

In sum, I would strongly advise the authors to discuss their work with a colleague who is highly familiar with cognitive biases in anxiety and particularly face processing biases.

Minor points:

- The hypotheses are stated in a much clearer way in the preregistration than they are in this manuscript. A point by point write-up may increase clarity
- The use of the cryptic task Acronyms has been changed to some degree but not consistently. They seem to come back in tables, figures and throughout the manuscript. The same counts for the 7.5%CO₂ vs Air descriptors.
- Some hints on literature that may help the authors in understanding the framework of cognitive processing in anxiety and potential use of trait vs state difference. There are more and also more recent ones, but that should give the authors an idea of what to look for ...

MacLeod, C., & Rutherford, E. (1992). Anxiety and the selective processing of emotional information: Mediating roles of awareness, trait and state variables, and personal relevance of stimulus materials. *Behaviour Research and Therapy*, 30(5), 479-491.

Schulz, S. M., Alpers, G. W., & Hofmann, S. G. (2008). Negative self-focused cognitions mediate the effect of trait social anxiety on state anxiety [Article]. *Behaviour Research and Therapy*, 46(4), 438-449. [://000255316200003](https://doi.org/10.1016/j.brat.2008.03.003)

Helzer, E. G., Connor-Smith, J. K., & Reed, M. A. (2009). Traits, states, and attentional gates: Temperament and threat relevance as predictors of attentional bias to social threat. *Anxiety, Stress & Coping*, 22(1), 57 - 76. <http://www.informaworld.com/10.1080/10615800802272244>

Reviewer: 3

Comments to the Author(s)

Dear authors,

Thank you for your work on how trait and state anxiety modulate emotional face processing. Considering that the literature is muddled in contradictory results, more robust, clear, and pre-registered studies such as this one are much needed. As other reviewers have already provided much feedback, I just have a few minor comments:

- Abstract: the authors should report all results for state anxiety and then for trait anxiety, as this would make for more logical description of the results. This would result in the following structure:

"The strongest emotion-specific effects were for happiness, with reduced accuracy ($p = .002$, $d_z = 0.49$) and sensitivity ($p = .004$, $\eta^2_p = .17$) during heightened state anxiety. However, there was evidence of increased anger bias and decreased happiness bias during heightened state anxiety, among individuals with high trait anxiety ($p = .03$)."

Furthermore, the “biases” should be phrased otherwise or explained within parentheses at some point of the abstract to clarify what is meant, e.g., “[state anxiety] appears to facilitate the processing of anger but impair the detection of happy expressions.” or something along those lines.

- The definition of trait and state anxiety deserves a sentence of its own, so this should be separated from the rest of the sentence: "are transient reactions to environmental stressors (31, 32)."

- Power calculations: the authors should report the power afforded by the current sample assuming a d of .69.

- Internal consistency estimates (preferably McDonald's Omega) should be reported for all measures of interest, including both questionnaire and task-based measures.

- All p -values should be reported to the third decimal place, following APA good practice recommendations.

- I am missing a discussion on the potential behavioral consequences of the observed biases. Without straying too far from the data, the authors should briefly discuss how the observed biases might play out in terms of real-life approach-avoidance behavior. This would also dovetail with the Introduction, where the authors do comment on these issues. Moreover, the authors should more precisely delimit the relevance of the reported effects for anxiety in comparison to other psychopathological dimensions, as similar biases have been reported for individuals high on trait anger and aggression.

Reviewer: 4

Comments to the Author(s)

The current manuscript compares the interactive influence of state and trait anxiety on facial expression recognition. The experimental manipulation of state anxiety (i.e, 7.5% carbon dioxide challenge) as well as the assessment of emotion recognition (biases) with 6AFC and 2AFC tasks is innovative and clearly improves our understanding of underlying psychological mechanisms. The manuscript in its current version demands a lot of inference from the reader and arguments are often not formulated clearly. I have some minor comments:

Abstract:

- As reviewer 1 already mentioned, it is unusual to report statistics in such an amount in the abstract as it decreases readability. If you really want such information in the abstract, please be consistent and report effect sizes to all p -values and use either two or three decimal places

- Reporting the place of recruitment in the abstract is also highly unusual and doesn't add valuable information in my opinion

- I don't think that "7.5% carbon dioxide challenge" is a widely known paradigm, maybe you can enter a short descriptive sentence about this experimental manipulation including its sham equivalent instead

Introduction:

- Last paragraph p4: "Measurement differences may also explain mixed findings. In general, accuracy reflects hits (correct identification of emotions), bias reflects hits and false alarms (incorrect identification of emotions), and sensitivity reflects hits whilst accounting for false

alarms.” □ Statistic might be a better word compared to measurement. This whole paragraph is quiet confusing and demands a lot of inference by the reader. Please clarify your arguments.

- P5L5: What is “naturally occurring state Anxiety”?

- P5L18: What is “7.5% carbon dioxide (CO₂) challenge”? Maybe you don’t have to define this paradigm here, but can describe the conceptual idea “participants inhale XXX to XXX “

Method:

- P7L39: “Participants were eligible if they met our criteria for low or high trait anxiety □ Which are? Please refer to the paragraph where you describe your criteria

- P8L10: “Scores between 21-31 and 44-64 denoted low and high trait anxiety, respectively. For the last two participants, we lowered the threshold of inclusion for high trait anxiety to 41-64 to support recruitment. □ Very complicated, maybe only: “Scores between 21-31 and 41-64 denoted low and high”?

- P9L26: Maybe you like to add the following reference which found that neutral and angry faces elicit comparable negative facial responses when they are passively viewed which indicates the negative valence of neutral faces: Höfling, T., Alpers, G. W., Gerdes, A. B. M., & Föhl, U. (2021). Automatic Facial Coding Versus Electromyography of Mimicked, Passive and Inhibited Facial Response to Emotional Faces. *Cognition and Emotion*, 35(5), 874-889.

<https://doi.org/10.1080/02699931.2021.1902786>

- P10L12: What is “A-prime”?

- ANOVA is a common abbreviation

Results:

- Again, varying decimal place of p-values, please unify

- “There was also strong evidence of a state anxiety emotion interaction [$F(3.76, 165.36) = 4.11, p = .004, \eta^2 = .09$]. □ This is surely not a large effect, please indicate your effect size interpretation in the corresponding method section

- The assumption of sphericity was violated for the emotion, and state anxiety × emotion within-subjects effects, so here we report Greenhouse Geisser statistics. □ Such information needs to be in the corresponding method section and not in the results section

- P16L33: non-significant differences of mean scores are interpreted. There is clearly no difference between them.

- Please move your manipulation check from the end to the beginning of the results (order of tables is currently wrong too)

- “gas inhalation order modified the effect of state anxiety on emotion recognition accuracy ($p = .01$), with stronger effects when the 7.5% CO₂ inhalation came first.” □ Is the robustness of your findings threatened? Please provide statistics and more details on this effect as a supplement

===PREPARING YOUR MANUSCRIPT===

===PREPARING YOUR REVISION IN SCHOLARONE===

<https://royalsociety.org/journals/authors/author-guidelines/#supplementary-material> to include a suitable title and informative caption. An example of appropriate titling and captioning may be found at https://figshare.com/articles/Table_S2_from_Is_there_a_trade-off_between_peak_performance_and_performance_breadth_across_temperatures_for_aerobic_scops_in_teleost_fishes_/3843624.

Author's Response to Decision Letter for (RSOS-210056.R1)

See Appendix B.

Decision letter (RSOS-210056.R2)

Dear Dr Dyer,

It is a pleasure to accept your manuscript entitled "The Role of State and Trait Anxiety in the Processing of Facial Expressions of Emotion" in its current form for publication in Royal Society Open Science. The comments of the reviewer(s) who reviewed your manuscript are included at the foot of this letter.

===COVID-SPECIFIC TEXT -- WILL ONLY BE ADDED TO COVID-PAPERS BY THE EDITORIAL OFFICE===

COVID-19 rapid publication process:

We are taking steps to expedite the publication of research relevant to the pandemic. If you wish, you can opt to have your paper published as soon as it is ready, rather than waiting for it to be published the scheduled Wednesday.

This means your paper will not be included in the weekly media round-up which the Society sends to journalists ahead of publication. However, it will still appear in the COVID-19 Publishing Collection which journalists will be directed to each week (<https://royalsocietypublishing.org/topic/special-collections/novel-coronavirus-outbreak>).

If you wish to have your paper considered for immediate publication, or to discuss further, please notify openscience_proofs@royalsociety.org and press@royalsociety.org when you respond to this email.

===END OF COVID-SPECIFIC TEXT -- WILL BE REMOVED AS NECESSARY BY THE EDITORIAL OFFICE===

on behalf of Dr Inti Brazil (Associate Editor) and Essi Viding (Subject Editor)
openscience@royalsociety.org

Associate Editor Comments to Author (Dr Inti Brazil):
Associate Editor
Comments to the Author:
(There are no comments.)

Reviewer comments to Author:

Response to Reviewers' Comments (RSOS-210056)

The Role of State and Trait Anxiety in the Processing of Facial Expressions of Emotion

Thank you to the editors of RSOS and the reviewers for their thoughtful and thorough comments that have helped us to improve our manuscript. Please see our responses below.

Reviewer 1

Abstract:

- 1. Even with a rigorous experimental set-up, I would be always cautious with regard to causal inferences.**

Changed - page 2: 'State anxiety appears to influence facial emotion processing...'

- 2. It is not clear that the challenge refers to an anxiety induction.**

Changed - page 2: 'High state anxiety, induced using the 7.5% carbon dioxide challenge...'

- 3. It is always helpful to report what kind of sample was recruited. Healthy sample recruited from general population? Male:female ratio? Students? High/low trait?**

Changed - page 2: 'Healthy participants (N = 48, 50% male, 50% high trait anxiety) were recruited from Bristol, UK.'

- 4. As far as I know it is unusual to report the statistics in the abstract.**

We have reported p-values and effect sizes to support results statements in line with recommended reporting checklists: <https://onlinelibrary.wiley.com/doi/full/10.1111/add.14269>

Introduction:

- 5. The introduction of the relevant theoretical framework is too shallow if not absent. The intro basically consists of short definitions of trait vs state anxiety and a quite thorough collection of studies pleading for or against emotion processing biases when anxiety is concerned. Yet, the whole framework/theoretical bases of processing biases in anxiety is neglected except for one mentioning on page 4, line 43. I think that the general idea of cognitive but at least that of face recognition biases and interpretation biases in particular should be explained in much more depth. What is important for example, is the fact that these biases make sense in specific situations, but are too prominent in anxiety disorders. What is also very important, is the fact that these biases are context and anxiety specific. For a spider phobic, a faster/more accurate recognition of facial expression makes no sense when in a situation of immanent 'spider-threat', nor does the negative interpretation of a happy face. For a person with social anxiety, faces are the cue to potential rejection and quick recognition or detection of such threat make a lot of sense. Here, misinterpretations of ambiguous faces may even increase state anxiety. The introduction pretty much relies on social anxiety literature, not questioning the relevance of face cues in un-social threat scenarios. Irrespective of the theoretical framework that should justify the choice for these particular processing aspects and the stimulus selection, the choice of state anxiety induction is also noteworthy. I have no doubts that the CO2 challenge evokes symptoms and distress associated with anxiety states. Yet, it is also a threat clearly based on aversive internal physiological symptoms clearly associated to panic disorder. Why should such an internal state have any effect on improved or biased**

threat-detection in the environment and particularly for faces when anxiety in general is at stake?

The introduction has been restructured and more discussion of and references to theoretical models have been included:

Added - page 3: 'Cognitive models suggest that anxiety is related to cognitive biases at the stage of initial processing (i.e., attention). According to Gray's Reinforcement Sensitivity Theory (8), the behavioural inhibition system predicts increased vigilance towards threat cues in anxiety. Preferential attention to threatening stimuli, such as facial expressions of negative emotions, may have adaptive value - for example, by discouraging potentially costly interactions. However, hypervigilance towards and difficulty disengaging from threatening stimuli are considered central to the aetiology and maintenance of anxiety (9, 10).'

Added - page 5: 'Williams and colleagues (39) distinguish between state and trait anxiety. According to their cognitive model, state anxiety influences the perception of threat (affective decision mechanism), and trait anxiety determines whether processing resources are directed towards (high trait anxiety) or away from (low trait anxiety) a stimulus perceived to be threatening (resource allocation mechanism). Therefore, high trait anxious individuals may have a greater attentional bias towards threat, whereas low trait anxious individuals may exhibit attentional avoidance.'

The case for the 7.5% CO₂ challenge has been extended.

Added - page 5: 'These findings support cognitive models that argue that state anxiety impairs emotion processing (37). The Clark and Wells (38) model of social phobia attributes this to a shift in attentional resources toward internal cues (i.e., anxiety symptoms) and away from external cues (i.e., facial expressions), which may also occur during the 7.5% CO₂ challenge.'

Note that we used the 7.5% dose rather than the 35% dose. The latter induces panic symptoms rather than increased state anxiety. In particular, the 35% CO₂ model (single breath) activates the hypothalamic-pituitary-adrenal axis, increasing adrenocorticotrophic hormone and cortisol levels (Argyropoulos et al., 2002; Kaye et al., 2004), whereas the 7.5% CO₂ model does not (Bailey, Argyropoulos, Lightman, & Nutt, 2003).

We have now explicitly stated that these biases are context- and anxiety-specific, and a limitation has been added to the discussion:

Added - page 20: 'Furthermore, biases in facial emotion processing are context- and anxiety-specific. Facial cues are clearly less relevant in non-social threat situations and certain anxiety disorders (e.g., specific phobias related to animals or environments). Therefore, there are limits to the generalisability (external validity) of these findings.'

6. There is also some confusion when recognition accuracy and sensitivity are introduced. The parallels/differences are not clear.

Added - page 4: 'Measurement differences may also explain mixed findings. In general, accuracy reflects hits (correct identification of emotions), bias reflects hits and false alarms (incorrect identification of emotions), and sensitivity reflects hits whilst accounting for false alarms.'

7. Considering the whole field of face processing (biases) it is also unclear why particularly the interpretation bias is chosen and why particularly that of anger vs happy. In the light of the general claims that are made concerning the influence of anxiety on face processing, this would make more sense when looking at social anxiety, but not necessarily for anxiety

in general. Wouldn't it have made more sense to investigate the general and emotion specific recognition differences per condition and contrast them with the sensitivity measures? Or if the interpretation biases can be theoretically linked contrast recognition and interpretation only and investigate whether recognition and biases (for specific emotions) are related?

Interpretation biases were chosen because as stated in the introduction, interpretation biases have been demonstrated in individuals with different types of anxiety, but evidence is more limited for state and trait anxiety. We previously found that 7.5% CO₂ induced anxiety (i.e., not specifically social anxiety) *does* have an effect on emotion processing. Facial expressions of emotion also have the potential to be threatening to everyone, whether they have an anxiety disorder or not. We have provided more justification for choice of emotions in the 2AFC:

Added - page 6: 'Angry–happy facial morphs were selected to measure bias because cognitive models suggest biased threat detection in anxiety (9), because previous stress-induction procedures have induced anger biases (45), and to ensure consistency with our previous experiment.'

We were constrained by the previous study and the limited timing of the inhalation (for safety reasons). If we had used different morphs, and found different results to our previous study, it would be unclear whether this could be attributed to task differences. Limitations of the 2AFC have been outlined in the discussion, for example:

Page 21: 'Due to the limited time available during the inhalation procedure, we could not include several 2AFC tasks. It would be useful for a future study to investigate the effects of state anxiety on interpretation biases to other emotions, to determine whether the results for happiness are unique.'

8. With regard to the hypotheses, I would suggest sorting them with regard to the process rather than anxiety type (state vs trait). That would also reflect the structure in the results section. In the light of the contradictory findings reviewed in the intro, the specific hypotheses are not intuitive. In addition, I would also mention the explorative part even without concrete hypothesis.

Hypotheses have now been separated by both anxiety type and emotion processing outcome to improve clarity (see also response to point 27), and explicit justification for the direction of these hypotheses have now been provided. We have also now re-ordered the hypotheses based on facial emotion processing outcome (recognition, bias) to mimic the structure of the results section. Secondary questions have been stated.

Changed - page 6: 'We hypothesised that high (a) state anxiety and (b) trait anxiety would lead to lower emotion recognition accuracy, and high (c) state anxiety and (d) trait anxiety would lead to increased interpretation bias for anger (and decreased bias for happiness), compared to low state and trait anxiety, respectively. In addition, we hypothesised that (e) the effects of high state anxiety on facial emotion processing would be greater among individuals who report high trait anxiety. State anxiety predictions were based on our previous findings, and trait anxiety predictions were based on the fact that high trait anxiety is characterised by increased frequency and intensity of state anxiety reactions than low trait anxiety (32) and previous studies showing altered emotion processing in trait anxiety. Finally, as hypothesis-free secondary analyses, we explored the roles of state and trait anxiety on emotion recognition accuracy and sensitivity to specific emotions. Angry–happy facial morphs were selected to measure bias because cognitive models have proposed biased threat detection in anxiety (9), previous stress-induction procedures have induced anger biases (45), and to ensure consistency with our previous experiment.'

Methods:

- 9. *The method seems thorough and accurate but the terminology is confusing. The authors stick to the not quite intuitive acronyms of the tasks they used instead of the concepts they measure. The same counts for the terminology with regard to the anxiety explanation/induction: After a first introduction, I think something like (induced) state anxiety vs control or high vs low state anxiety would make the text much more readable.***

The state anxiety manipulation has been clarified, and all references to gas (air versus 7.5% CO₂) have been changed to state anxiety (low versus high).

Changed - page 7: 'For the 2AFC and 6AFC tasks, there was a within-subjects factor of state anxiety (low, high), corresponding to medical air and 7.5% CO₂ enriched air conditions, respectively...'

References to threshold scores (2AFC) have been changed to 'interpretation bias.' References to 6AFC hits have been changed to 'emotion recognition accuracy.'

- 10. *I wonder if there was a particular reason to not include 'neutral' in the basic set and as to-be-recognized- expression. Of course this is not an emotion but it would (a) allow to contrast emotions and non-emotion recognition directly, (b) dilute the set of primarily negative emotions and (c) may allow to identify recognition/interpretation biases in one go, e.g., by identifying what people see if the 5% emotional signal is present and neutral is an option.***

One key feature of the stimulus sets we have developed lies in the construction of the stimuli themselves. We employ composite faces generated from a larger number of individual photographic subjects. This well-established technique isolates the prototypical characteristics of emotional expressions, while removing the idiosyncratic variation in expression that is found between individuals. This is principally because of recent evidence which suggests that visual representations of emotion are better described as being coded with reference to a prototype of this sort, as opposed to a neutral face (Skinner & Benton, 2008).

Another feature is how these stimuli are then used to generate morph sequences. Many tasks typically employ sequences that run from a neutral exemplar to an emotional exemplar (e.g., neutral to happy). Instead, we have generated continua that start from an average or prototypical emotional face, constructed by compositing exemplars of each of the six basic emotions (i.e., anger, happiness, sadness, fear, surprise and disgust). This face appears genuinely emotionally ambiguous, rather than neutral. This is because a neutral face is not actually without emotional content; some subgroups of participants, such as those with high levels of anxiety, appear to default to interpreting neutral faces as threatening (Yoon & Zinbarg, 2008).

Extended - page 9: 'There is evidence that an emotional prototype face is likely to be a better approximation of the centre of emotional 'face space' than a neutral face (50) and a neutral face is not without emotion.'

- 11. *I'm curious if response times are assessed and if the researchers have looked at speed-accuracy trade-offs. Maybe this trade-off maybe something that changes under stress/anxiety.***

Response times were recorded by default on E-Prime. However, the effect on speed is a different research question which was beyond the scope of the current study. Therefore, we did not extract this information from the raw E-Prime data files.

Added - page 21: 'Finally, future studies should also examine the role of state and trait anxiety in the speed of facial emotion processing, which may help to elucidate these findings. For example, there is evidence that socially anxious individuals are faster at detecting facial expressions of emotion at moderate (versus low and high) levels of anxious arousal (37).'

12. I still find the choice for Happy-angry interpretation bias somewhat arbitrary and not convincingly theoretically founded. More combinations or morphs of all emotions with neutral may have shed a clearer picture.

Please see response to point 7.

13. With regard to prescreening and recruitment I wonder whether this study is based on an own sample or if it is part of a bigger dataset and larger study population. Since procedure and set-up are quite similar to the studies from their own lab the authors seek to replicate and repeatedly cite, this is hard to disentangle. To be clear: It is no problem to seek to publish different subsets of a bigger study, but transparency must be warranted.

Different participants were recruited for this study. This was not a subset of a previous larger study.

14. What was the exact setup of the VAS scales, what were the questions asked and what were the anchors? The MINI is a semi-structured diagnostic interview. To my knowledge it is not officially translated to a self-report version.

Details of all questionnaire items are reported in the data dictionary which will be made available to readers on the University of Bristol Repository. To improve transparency, these have been added to the manuscript as well.

Added - page 8: 'The VAS had 11 items (alert, sedated, fearful, relaxed, anxious, happy, feel like leaving, tense, nervous, worried, stressed) from 0 (not at all) to 100 (extremely).'

We used the term 'self-report' to cover methods that gather subjective responses from participants, to contrast with the methods that objectively measure other criteria. Self-report methods include questionnaires and interviews. We have now clarified that we use a truncated version of the MINI that is used for screening.

Changed - page 11: 'All other criteria, including psychiatric health (using a screening tool adapted from the Mini-International Neuropsychiatric Interview) (51) were assessed by self-report.'

15. With regard to the inhalations it is not clear whether participants wear the masks/inhale throughout the whole time or only at the beginning of the task and at what point the state measures took place.

Changed - page 11: 'Participants were fitted with an oro-nasal mask. They inhaled the gas (air or 7.5% CO₂) for one minute to allow anxiety levels to stabilise, then they completed both computer tasks (6AFC, 2AFC) while continuing to inhale the gas. Inhalations lasted up to 20 minutes. Immediately after the inhalation, masks were removed, HR and BP were measured again and participants completed the STAI-State, PANAS, VAS (reporting on how they felt during the inhalation when the effects were at their peak).'

Results:

16. The authors should consider putting the means of the participant characteristics in the same table as the comparisons of the state measures (table 3?) rather than in the text. In addition, they should also statistically compare the general characteristics between groups

to verify that the stratification with regard to trait has not brought along any other unwanted group differences except the expected ones.

We have conducted t-tests to compare participant characteristics between trait anxiety groups. These results have been added to the table which reports differences between state anxiety measures. Tables have been re-ordered to reflect their new positions in the results section. We have decided to keep the description of participant characteristics for the whole sample within the text rather than in a table, due to space constraints.

Added - page 14: 'Participant characteristics between groups were similar, except that anxiety sensitivity and neuroticism were higher and extraversion was lower in the high (versus low) trait anxiety group (Table 1).'

17. I would also structure the results in line with the mentioning of the tasks. Up until here the recognition task is always mentioned first and then the interpretation task is mentioned. In the results section it is the other way around. Besides that, as mentioned in the intro, it would make sense to sort the hypothesis by task rather than by anxiety-type. That would also give the manuscript a much clearer structure.

To keep the order of tasks consistent, we now refer to the 6AFC before the 2AFC in the statistical analyses section of the methods (page 12), and the results section (page 14). The hypotheses are now ordered to match the structure of the results section (page 6).

18. The authors talk about 'some evidence' for 'significant results and 'no clear evidence' for non-significant findings even when far off non-significant 'trends'. They also tend to interpret the differences in the mean scores in non-significant findings. Despite being incorrect, this framing of the results is also misleading. Please, talk about, eg., significant differences vs no differences, effect vs no effect, or something alike. The description of the high-state anxiety condition as 'gas (i.e., state anxiety)' is confusing. Please, consider comprehensive rephrasing here and throughout the manuscript.

As per reporting guidelines (West et al., 2018) we have refrained from reporting 'no difference' between conditions because we have not demonstrated that through, for example, Bayes factors. Instead, we consider language that refers to 'evidence of a difference' to be more appropriate here. Furthermore, we do not agree with dichotomisation of results, into 'significant' or 'non-significant' according to an arbitrary p-value threshold, for reasons reported elsewhere (Sterne & Davey Smith, 2001).

We think it is appropriate to comment on the direction of the differences for findings related to a directional hypothesis e.g., page 9: 'Although scores were lower in the high state anxiety (M = 6.78, SD = 1.41) than the low state anxiety (M = 7.07, SD = 1.14) condition, there was no clear statistical evidence of a difference.'

As per point 9, all references to gas (air versus 7.5% CO₂) have been changed to state anxiety (low versus high).

19. Also should the statistical results be translated to 'understandable' language without interpreting them here in the results: e.g., 'there was some evidence of a difference in threshold scores between low vs high trait anxiety scores'. Does that means something like: 'there was a tendency in the high trait anxiety group to interpret happy as angry'?

We are not clear which finding the reviewer is referring to here, as this is not a direct quote from the paper. We have reported on page 15 that there was no main effect of trait anxiety. However, we have rephrased the interaction results (which we think they may be referring to) into more intuitive language.

Page 15: 'In the high trait anxiety group, there was some evidence of a difference in interpretation bias between the low and high state anxiety conditions (7.29 vs. 6.63, $p = .03$), indicating greater biases towards perceiving anger when experiencing high state anxiety, but there was no evidence in the low trait anxiety group (6.82 vs. 6.95, $p = .59$).'

20. The paragraph about the 6AFC starts with indicating a 2x2 model while the first result presented is a main effect. That is confusing.

Indeed, in the statistical analyses section we report that we ran a 2 x 2 model in SPSS; however, it is typical, and it was of interest, to report the main effects of each independent variable before reporting the interaction.

21. The authors should consider (most recent) APA norms for reporting results and take into account when the zero before a decimal point is reported and when not.

Leading zeros have been added to F-values, and Cohen's d values where they were missing. However, we understand that the Royal Society supports format-free initial submission (<https://royalsociety.org/journals/authors/author-guidelines/#formatting>).

22. On page 14 line 56 they talk about a smaller effect size while indicating earlier that it was strong evidence. I feel that the evaluation of how strong a particular effect is should be done in the discussion section. Here, it should be merely reported.

Reference to the effect size being smaller has been removed from the results section, and instead included in the discussion (page 17).

23. To me it appears that the evaluation of fewer hits 'particularly' for happiness may result from a comparably subjective evaluation if effect-sizes are not taken into account.

We are unclear what the reviewer means by this point. The term 'particularly' was used precisely because the effect size was largest, and the p-value was smallest for happiness relative to the other emotions (Table 2). These were objective comparisons.

24. The sensitivity results should be marked more clearly as exploratory here.

We planned to examine emotion recognition sensitivity (secondary rather than exploratory analyses). However, the methods we used were different to what we stated in our preregistered study protocol on the Open Science Framework. We have been transparent about this in our methods section.

25. The manipulation check is a valuable addition. Here, the participant stratification could be mentioned as well, if not done earlier. It would also be valuable to report the results of 'counterbalancing' analyses here. Did the order of anxiety manipulation and/or tasks make a difference?

Participant stratification results have now been reported on page 14 (see point 16 above).

We have examined whether the counterbalancing order influenced the results:

Added - page 16: 'There was no clear evidence that gas inhalation order or task order modified the effects of state and trait anxiety on interpretation bias for anger ($p > .25$). However, gas inhalation order modified the effect of state anxiety on emotion recognition accuracy ($p = .01$), with stronger effects when the 7.5% CO₂ inhalation came first.'

Discussion:

26. In general, I find the discussion (as the introduction) shallow and confusing with regard to straightforward terminology and theoretical purpose. Facial emotion processing in the context of anxiety is used too broad when only few aspects are taken into account.

Please see responses to other related points. We have now embedded more theory into our introduction and discussion, made the terminology more straightforward, and highlighted the generalisability limitations of our findings given that this was just one study examining the role of specific types of anxiety.

27. It is confusing that the authors talk of their first hypothesis being confirmed while later on the say it is not. Maybe they should consider separating the hypotheses per task (as suggested earlier) and structure their results and discussion section accordingly.

We have now separated the 2 hypotheses into 4 hypotheses to distinguish between state and trait anxiety and the two tasks and referred to them individually in the discussion. We think that this has improved clarity.

28. On p16 lines 12ff, non-significant differences of mean scores are interpreted.

We think that it is important to comment on the direction of effect (point estimate) as well as the strength of the evidence against the null hypothesis (p -value) for results where we had an *a priori* directional hypothesis.

29. On p16 lines 33ff, the conclusion is rather far-fetched considering the fact that only one emotion pair was tested. I strongly suggest to strive for more theoretical imbedding of the results.

We have tempered the causal language and specified that only one emotion pair was tested, and we have related our findings to theoretical models.

Page 16: 'Post hoc tests indicated an increased tendency to perceive anger and a decreased tendency to perceive happiness in the high (versus low) state anxiety condition, among individuals with high (but not low) trait anxiety. In other words, a situational spike in anxiety appeared to cause a greater anger bias in emotion processing for individuals with a dispositional tendency to experience anxiety. These findings support cognitive models which propose that different patterns of bias in high and low trait anxious individuals become more pronounced as stimulus threat value or state anxiety increases (39). Although results should still be interpreted with caution, as only angry-happy facial morphs were included.'

30. Explanation of the results are primarily sought in technical and methodological differences rather than in underlying mechanisms and theoretical predictions. It seems at times that primarily earlier work of the authors is taken as reference point rather than any theoretical framework. The fact, for instance, that no main effect of trait anxiety was detected makes a lot of theoretical sense: It is a latent trait, and it has been argued that the associated processing patterns are only activated when the (more frequently) occurring anxiety states are triggered in the high-trait anxious individuals. But again,

anxiety inducing suffocation signals may not be directly related to processes going on in fears of negative evaluation, fear of spiders, etc. On the other hand, threatening facial expressions are not necessarily relevant in situations when one fears to suffocate. These aspects should be disentangled.

Added – page 17: ‘These findings support cognitive models which propose that different patterns of bias in high and low trait anxious individuals become more pronounced as stimulus threat value or state anxiety increases (39).’

Added – page 19: ‘Indeed, some cognitive theories suggest that biases in emotion processing may in turn elicit autonomic arousal and sustain anxious states (62) and biased interpretation of ambiguous social cues is considered a maintenance factor for social anxiety disorder (63).’

Added – page 20: ‘Although the 7.5% CO₂ challenge is a well-validated human experimental model of anxiety (42, 43), it may not be directly related to social threat situations, where facial emotion processing is most relevant. It would be useful for future studies to examine these questions using state social anxiety manipulations (e.g., via the Trier Social Stress Test). Furthermore, biases in facial emotion processing are context- and anxiety-specific. Facial cues are clearly less relevant in non-social threat situations and certain anxiety disorders (e.g., specific phobias related to animals or environments). Therefore, there are limits to the generalisability (external validity) of these findings.’

Participants do not subjectively report feelings or fears of suffocation. The sensation is similar to post-exercise breathing. People feel out of breath but can compensate (i.e., they may have to breathe more rapidly/deeply) but they do not feel unable to breathe or restricted of oxygen. While there are different mechanistic explanations of this, crudely hypercapnia turns on the anxiety response to potential respiratory threat. So, it is an anxiety response to an identified physical threat, but it is not suffocation or hypoxia.

In addition, the authors could, e.g., discuss if their ‘happiness’ findings couldn’t be related to the fact that happy is actually their only positive signal in the stimulus set. It could appear as the ‘odd-one out’ or are in general primed with a negative mindset especially in a threatening situation influencing their choices in the tasks.

We have been more cautious in our interpretation of emotion-specific effects (specifically for happiness), given that these were hypotheses-free secondary analyses. We have removed reference to the happiness findings from the conclusions section of the discussion and the abstract and instead focused on the results for which we had hypotheses.

Abstract: ‘State anxiety appears to impair emotion recognition accuracy, and among individuals with high trait anxiety, it appears to increase anger (and decrease happiness) biases. Trait anxiety alone does not appear to be associated with facial emotion processing.’

Page 19: ‘These findings are also interesting given that happy facial expressions are reported to be more easily identified than negative facial expressions (64). However, we had no *a priori* hypotheses for specific emotions, therefore strong conclusions cannot be drawn from our data. The effects of anxiety on the processing of happy facial expressions, and the mechanisms behind a possible impairment, should be specifically investigated in future studies. For example, deficits could be related to the fact that happiness was the only positive signal in the stimulus set and induction of negative affect could lead people to attend to negative information.’

The methodology is very interesting and surely has potential to be useful for anxiety research, but with a lacking theoretical bases the assumption that the results may be relevant for understanding anxiety disorders is unfortunately far-fetched. Technical issues:

31. The paper could be more structured and in more depth.

The paper has been restructured according to recommendations, and further theoretical models have been incorporated.

32. The style of writing is fine but terminology could be more straight-forward.

Terminology has been simplified throughout.

33. Several APA errors with regard to reporting statistics are observed.

Leading zeros have been added to F-values, and Cohen's d values where they were missing. However, we understand that the Royal Society supports format-free initial submission (<https://royalsociety.org/journals/authors/author-guidelines/#formatting>).

34. Table 2 could be more condensed.

We do not think that Table 2 (now Table 3) can be condensed in any way. All results have been reported for transparency.

35. Table 3 could become part of a Table 1 in which the population descriptive means per group are depicted as well as those of the state measures. The results of difference testing could be added there as well.

The original Table 3 has been changed to Table 1 and differences between trait anxiety groups and state anxiety experimental conditions have both now been reported here.

In sum, I have my doubts that the manuscript should be published in its current state. Theoretical bases and clearer terminology should be provided to justify the choices made for the current set-up. In the end the results should be discussed in the light of the theories again, to show in how far they have increased our insight in the mechanisms of anxiety and potential impact for the clinical field.

Reviewer: 2

1. The used questionnaires are mentioned only briefly. I suspect more information can be found in the study protocol. This is fine but it would be transparent to report the psychometric properties, at least the reliability, of the tests used in the current study.

Added – page 8: 'The questionnaires are reliable and valid measures of the constructs they were intended to assess (49-52).'

Furthermore, unpublished research from a PhD student in our group (Maren Muller-Glodde) suggests that the 6AFC is a valid and moderately reliable measure of emotion recognition. The preprint of this study will be available on bioRxiv in the near future.

2. In the method section, page 10, line 47, abbreviations BP and HR are used. Even though these abbreviations are well known, they need to be written out/introduced.

Blood pressure (BP) and heart rate (HR) acronyms were defined on page 7 when they were first introduced.

- 3. In the method section, page 12, line 44-47, it is stated that a previously proposed analysis appeared to be insufficient. Therefore, 6 separate 2 × 2 models were examined. I am wondering whether the authors applied any correction for multiple testing? As there were quite some statistical analyses conducted in a relatively small sample, this should have been done.**

We did not apply a correction for multiple testing (e.g., Bonferroni) because these analyses were exploratory, and because we are not focused on the p-values and dichotomous significance testing, for reasons reported elsewhere (Sterne & Davey Smith, 2001). As per our response to R1, we have now interpreted these exploratory emotion-specific results more cautiously.

- 4. Throughout the manuscript, the interaction effects are indicated by using “x” instead of a multiplication sign. E.g, gas x trait anxiety instead of gas × trait anxiety**

This has been corrected throughout the manuscript.

- 5. In the discussion, page 16, line 12 – 18, the authors state that there was no clear evidence of a difference in interpretation bias but they interpreted the direction of the results, which was in correspondence with their expectations. However, the effect was statistically not significant. This means that also no trends can be observed and interpreted.**

Please see response to Point 28 from Reviewer 1.

- 6. No comment just a thought out of curiosity. In the literature about hostility biases, it has been suggested that aggressive individuals experience difficulties in processing social/emotional information because it is inconsistent with their (cognitive) schemas. They need more time to process schema inconsistent information because it differs from their expectations. Furthermore, due to a high emotionality, they may experience any more difficulties to assess the situation from different perspectives. Resulting in relying more on existing schemas. In turn, this makes to interpretation of social information in a hostile manner more likely. I was wondering whether such a mechanism could also be present in the case of anxiety? It would be interesting to discover whether such (or other) underlying mechanisms apply to biases in social/emotional information processing across psychopathologies.**

This is an interesting question and a plausible mechanism. However, we have not examined the possible influence of schemas. We are not aware of experimental manipulations for other symptoms (except perhaps depression). Mendelian randomisation could be a method used to explore the causal effects between genetic liability for a psychiatric condition and facial emotion processing (<https://www.medrxiv.org/content/10.1101/2021.05.06.21256771v1>).

Additional changes to the manuscript:

To compensate for the additional text added in response to reviewer comments, we have shortened the manuscript slightly in other places, by making sentences more succinct.

We have added another study citation on page 5: ‘Whereas another study found high state anxiety impaired emotion recognition more for people with high (versus low) trait anxiety (40).’

Terminology has been simplified in the figures.

We have added to the acknowledgements and competing interest statements (apologies this was missed).

Response to Reviewers' Comments (RSOS-210056) – Revision 2

The Role of State and Trait Anxiety in the Processing of Facial Expressions of Emotion

Associate Editor Comments to Author (Dr Inti Brazil):

The reviewers from the previous round were mixed in their re-evaluation of the revised manuscript. I decided to secure additional reviews from other experts. As you will see, multiple reviewers highlight that there are still major conceptual issues and that the theoretical embedding is not sufficiently developed. One of the reviewers found it particularly difficult to follow the line of reasoning, as it requires a lot of inferencing from the reader. There are also methodological points that require attention. I believe that the comments are clearly articulated and can be used to improve the manuscript.

Thank you to Dr Brazil and the editorial team at RSOS for the opportunity to submit a second revision and thank you to the reviewers for taking the time to evaluate our manuscript. This detailed and constructive feedback has helped us to further improve our work. We have now developed the embedding of theory into the manuscript, clarified the line of reasoning, and addressed the methodological points. Please see below for details; page numbers refer to the tracked changed version of the manuscript.

Reviewer 1:

- 1. By revising their manuscript the authors have improved their manuscript considerably with regard to structure and readability. Yet, I am still not convinced about the theoretical embedding of the study. In particular the authors do introduce two models in the introduction (Gray as well as Williams et al.) Both models suggest that state as well as trait anxiety should improve or prioritize the processing of fear relevant cues. The fact that certain traits lead to overexaggerated (rapid detection of supposed threat, or seeing threat where is none, difficulty disengaging) nicely fit in that idea. Even quick recognition of threat makes sense here, as it may be that hyper alertness increases perceptual vigilance. You could say that anxiety pathology seems to be an over-sensitivity of potential threat detection and its processing. While quick processing of real threat seems to have a possible evolutionary advantage, it may be disadvantageous when specific anxious traits may bias an organism in preferentially 'looking' for the feared object in question (leading to chronic anxiety related stress and increases in biased processing). Thus from an evolutionary AND from a psychopathological standpoint preferential processing of (potentially) threatening stimuli make sense.***

We have now further developed the theoretical embedding of the study and incorporated your suggested text into the manuscript.

Changed – page 3:

‘Cognitive biases in emotion processing are common in anxiety (8). People with anxiety disorders, who typically exhibit heightened trait and state anxiety, are characterised by processing biases towards emotionally threatening stimuli (9). Cognitive models suggest that anxiety is related to cognitive biases at the stage of initial processing (i.e., attention). According to Gray’s Reinforcement Sensitivity Theory (10), the behavioural inhibition system promotes increased vigilance towards threat cues in anxiety. Preferential attention to and quick processing of real threat may confer a

possible evolutionary advantage. For example, attention to facial expressions of negative emotions, may have adaptive value by discouraging potentially costly interactions. However, excessive sensitivity to potential threat detection and its processing, which appears to operate in anxiety pathology, may be disadvantageous when an individual is biased to preferentially ‘look’ for feared stimuli. Indeed, hypervigilance towards and difficulty disengaging from threatening stimuli are thought to be central to the aetiology and maintenance of anxiety (11, 12), as pharmacological interventions are associated with reductions in negative cognitive biases (13). Therefore, a generally adaptive system of preferential processing of potential threats can become maladaptive when dysregulated (i.e., in anxiety disorders).’

- 2. *In my eyes, the contradictory (previous) findings (of the authors) need to be seen in the light of these models rather than merely summing them up. The mentioned meta-analysis of Demenescu, for instance DOES indicate a moderate deficit of face recognition in anxiety disorders, BUT they do discuss these findings in the light of the models in the sense that advantages of threat processing in one domain (eg. vigilance/quick detection) may come at the cost of deficits in another domain (eg. overgeneralization/interpretation of negative as threat or erroneous categorization/recognition). To my knowledge all articles that present conflicting results do at least some attempts to explain them in the light of the current/dominant theories or try to nuance them by suggesting alternative theoretical explanations. The current paper does not do that.***

We have added some discussion of the findings in relation to the cognitive models. There is a considerable variability in findings across many studies, and ours is not the first to report null or opposite findings. Theory needs to be based on solid, replicable findings, rather than revised every time a null or opposite result is reported. We are reporting a study that uses very similar methods to our previous work and replicated some – but not all – previously reported findings. As such we are contributing to the weight of evidence from which solid theory can be developed – a point we return to below. Instead of suggesting alternative theoretical explanations for these findings, we acknowledge later in the limitations section possible problems with our 2AFC task (which measures interpretation bias towards perceiving anger) and our measure of trait anxiety. We have also now commented on the analyses of internal consistency reliability, which may also account for the contradictory findings.

Changed – page 19:

‘Contrary to our third hypothesis and the previous study, interpretation bias towards perceiving anger did not appear to differ under high and low state anxiety conditions. Furthermore, we did not find evidence to indicate a relationship between trait anxiety and interpretation bias towards perceiving anger. Therefore, these findings are not consistent with cognitive theories which suggest greater attentional bias towards threat in anxiety (10, 42) or previous evidence of an anger emotion processing bias in anxiety (5, 14, 28). *Post hoc* analyses suggest that the measurements obtained from the 2AFC task (interpretation bias towards perceiving anger) had moderate reliability as measured by internal consistency. As stated by Parsons and colleagues (67), ‘reliability is estimated from the scores obtained with a particular task performed by a particular sample under specific circumstances.’ Therefore, although the task and the method of anxiety induction were consistent across the current and previous study, it is possible that the samples differed in some way, which may account for this lack of replication. Nonetheless, there are no *a priori* reasons to suspect that this sample is systematically different from the previous study, and so the robustness of our earlier finding is questionable.’

In the next paragraph, we discussed the interaction results in relation to theory.

Page 20:

‘These findings support cognitive models which propose that different patterns of bias in high and low trait anxious individuals become more pronounced as stimulus threat value or state anxiety increases (42).’

Other contradictory findings have also been elaborated on.

Added – page 20:

‘These findings largely support the previous study, although Attwood and colleagues (38) observed a deficit in recognition accuracy for anger (Study 1) and surprise (Study 2), in addition to happiness, disgust, and fear. Discrepancies like these are common in this field. However, because we have used similar methods in both studies, this suggests that the effects may be transient or possible sample differences may explain this discrepancy.’

We had suggested alternative theoretical explanations elsewhere in the discussion. However, we are wary of *post hoc* rationalisation and theorising. There are many contradictory findings in the literature, and we instead have presented our results in a relatively neutral way, in an attempt to build a more solid evidence base to support subsequent theory development. It is important to find robust effects through the use of high-quality measures, rigorous experimentation, and reporting of all results.

Added – page 23:

‘Our findings may also have implications for theory and research. First, the consistency across our measures (accuracy, sensitivity, and bias) for happiness but not anger, suggests that alterations in the processing of positive (versus negative) emotions may play a more important role in the cognitive aspects of anxiety, a view echoed by other researchers (26). However, a more solid evidence base is needed to support subsequent theory development.’

- 3. *In addition, the suggestion that erroneous face processing may lead to problems in social interaction is, apart from socially anxious individuals, hardly the problem. The current study should primarily increase mechanistic or maybe methodological understanding of threat processing in anxiety. [E.g., is it possible that the method used is very reliable in inducing state anxiety or ‘symptoms’ related to it but that the potential changes in CO2 may have a physiological impact on cognitive processing irrigated to fear? Could it be possible that lab induced physiological fear states that are with no respect related to the task at hand are fundamentally different from fear states used in experimental work such as ‘announcements of a speech to be held after the task (for socially anxious individuals)’ even though physiological measures may be identical? What would that mean for our theoretical and mechanistically understanding of state anxiety across the different anxiety disorders and in general?] I am convinced that the authors have a solid knowledge of experimental and statistical techniques striving for useful and necessary replication of their earlier work but I cannot help the notion that they are either not aware of or are somewhat negligent with thoroughly seeking out the theoretical basis for their project, their hypotheses and implications of their results.***

Added – page 22:

'The 7.5% CO₂ inhalation induces physiological and psychological symptoms akin to generalised anxiety disorder (GAD) (45), increasing self-reported state anxiety, HR, BP, and hypervigilance to threat (45-47). The CO₂ inhalation may have an impact on cognitive processing related to threat. This is supported by Garner and colleagues (47), who found that it modulates attention mechanisms (i.e., alerting and orienting) involved in the temporal detection and spatial location of salient stimuli. From an experimental perspective, there are benefits of using the 7.5% CO₂ inhalation model. First, it yields a reliable unconditioned anxiety response that is less susceptible to individual variation (e.g., compared to models that incite conditioned anxiety responses). Second, unlike some models that induce anxiety and subsequently measure the outcome of interest, the tasks are completed during peak anxiety induction. However, there are different types of anxiety manipulation, and it is possible that experimental tasks that involve a social component (e.g., public speaking) may have different cognitive effects compared to the 7.5% CO₂ inhalation, which does not.'

- 4. *As I read the introduction the authors are not very precise in defining and distinguishing the biases they want to investigate particularly with regard to facial expressions. Recognition, sensitivity and interpretation are neither discussed separately nor is their possible interconnectedness mentioned.***

This paragraph has been expanded on to clarify the differences in the definitions of these terms and their interconnections.

Changed – page 5:

'Differences in the measures of facial emotion processing between studies may also explain mixed findings. In short, emotion recognition accuracy may be measured by hit rate (e.g., the correct identification of anger if angry faces are presented). If an individual demonstrates a higher hit rate for anger, this suggests that they have superior emotion recognition accuracy for anger. However, this measure of emotion recognition accuracy does not account for the times when an individual identifies anger in faces that are not angry. These are known as false alarms/errors (i.e., the incorrect identification of the emotions presented). A bias towards making angry responses may manifest in a higher hit rate and a higher false alarm rate, whereas sensitivity reflects hit rate whilst accounting for false alarms. An additional complication is that the term 'bias' can also refer to different things (e.g., neutral or ambiguous emotional facial expressions interpreted as angry) depending on the task and stimuli used in a study. Furthermore, different statistical analyses can be used to measure both bias and sensitivity.'

The terminology throughout the manuscript gives more evidence of this omission: what is an increased bias for anger – is an angry expression seen as MORE angry than it actually is? How can a bias of happy be decreased? At what point is the absence of any bias established in this work?

In the context of our study, and the task we used to measure bias (2AFC), bias refers to 'increased interpretation bias towards perceiving anger (and decreased bias towards perceiving happiness).' This has now been defined fully in the hypothesis section on page 7.

As described in the methods, the 2AFC stimuli were 15-image linear morph sequences ranging in equally spaced emotional intensities from the full intensity happy exemplar to the full intensity angry exemplar from the 6AFC (Figure 2). Each image between the two full intensity images contained a proportion of both emotions (e.g., 90% happiness contained 10% anger). Therefore, a bias in one direction (e.g., towards anger) at the same time reflects a bias *away* from the other

direction (e.g., towards happiness). For this task, we measure relative bias across individuals. We have added some clarification to the methods section.

Changed – page 12: ‘As the continuum ranged from happy (image 1) to angry (image 15), lower threshold scores indicated greater interpretation biases towards perceiving anger (i.e., individuals show an earlier change from perceiving happiness to anger) relative to higher threshold scores. In other words, relatively more morphed faces on the continuum (that contain a proportion of happiness and anger) are perceived to be angry than happy.’

In the design section the authors speak of ‘emotional bias’ and later (in the discussion) of an anger emotion processing bias and an anger bias. Also in the discussion, they talk of three measures of facial emotion processing: accuracy, sensitivity and bias, again showing quite some indifference with regard to the nuances. In the field of (biased) face processing, accuracy and sensitivity can also relate to, eg. Probe detection in relation to threat faces detection or location of specific expressions in a grit of faces. Bias can relate to literally any aspect of face processing. In sum, I would strongly advice the authors to discuss their work with a colleague who is highly familiar with cognitive biases in anxiety and particularly face processing biases.

We have now kept the terms consistent for each task (i.e., ‘emotion recognition accuracy’ for the 6AFC task [unless referring to a specific emotion] and ‘interpretation bias towards perceiving anger’ for the 2AFC task).

We hope that we have now clarified the nuances in these three measures in the introduction section (please see our response point 4 above). In the discussion we have added the following:

Added – page 22:

‘However, we acknowledge that accuracy, bias, and sensitivity are complex constructs in the field of facial emotion processing, and they can be measured in several different ways.’

Minor points:

- 1. The hypotheses are stated in a much clearer way in the preregistration than they are in this manuscript. A point by point write-up may increase clarity***

Following feedback from reviewers in the previous round of revisions, hypotheses were separated by both anxiety type and emotion processing outcome to improve clarity, and to reflect the structure of the results section. We have now separated this list into shorter sentences to increase clarity.

Changed – page 7:

‘We hypothesised that (a) high state anxiety and (b) high trait anxiety would lead to lower emotion recognition accuracy, compared to low state and trait anxiety, respectively. We also hypothesised that (c) high state anxiety and (d) high trait anxiety would lead to increased interpretation bias towards perceiving anger (and decreased bias towards perceiving happiness), compared to low state and trait anxiety, respectively. In addition, we hypothesised that (e) the effects of high state anxiety on facial emotion processing would be greater among individuals who report high trait anxiety.’

- 2. The use of the cryptic task Acronyms has been changed to some degree but not consistently. They seem to come back in tables, figures and throughout the manuscript. The same counts for the 7.5%CO2 vs Air descriptors.***

References to 2AFC and 6AFC outcomes/data have now been changed to interpretation bias towards perceiving anger and emotion recognition accuracy, respectively. We have kept the terms 2AFC and 6AFC when we are referring to the tasks, specifically.

References to 7.5% CO₂ and air conditions have now been changed to high and low state anxiety conditions, respectively. We have kept the terms 7.5% CO₂ and air when referring to these as inhalations or gases, specifically.

- 3. Some hints on literature that may help the authors in understanding the framework of cognitive processing in anxiety and potential use of trait vs state difference. There are more and also more recent ones, but that should give the authors an idea of what to look for ...**

MacLeod, C., & Rutherford, E. (1992). Anxiety and the selective processing of emotional information: Mediating roles of awareness, trait and state variables, and personal relevance of stimulus materials. *Behaviour Research and Therapy*, 30(5), 479-491.

Schulz, S. M., Alpers, G. W., & Hofmann, S. G. (2008). Negative self-focused cognitions mediate the effect of trait social anxiety on state anxiety [Article]. *Behaviour Research and Therapy*, 46(4), 438-449. [://000255316200003](https://doi.org/10.1016/j.brat.2008.03.003)

Helzer, E. G., Connor-Smith, J. K., & Reed, M. A. (2009). Traits, states, and attentional gates: Temperament and threat relevance as predictors of attentional bias to social threat. *Anxiety, Stress & Coping*, 22(1), 57 - 76. <http://www.informaworld.com/10.1080/10615800802272244>

Thank you for the suggested reading.

Added – page 6:

‘Research by Macleod and Rutherford (9) supports this theory. They found that for individuals with high trait anxiety, state anxiety elicits a selective processing bias favouring threat related information (colour naming of words). Whereas, for individuals with low trait anxiety, state anxiety elicits a processing disadvantage for this threat related information.’

Reviewer 3:

- 1. Abstract: the authors should report all results for state anxiety and then for trait anxiety, as this would make for more logical description of the results. This would result in the following structure: “The strongest emotion-specific effects were for happiness, with reduced accuracy ($p = .002$, $d_z = 0.49$) and sensitivity ($p = .004$, $\eta^2 = .17$) during heightened state anxiety. However, there was evidence of increased anger bias and decreased happiness bias during heightened state anxiety, among individuals with high trait anxiety ($p = .03$).”**

The reason why we had originally placed the emotion-specific effects last was because these were hypothesis-free secondary analyses. For clarity, and due to word count limitations with the other abstract additions requested, we have now removed secondary analyses from the abstract. Now, only the main results and conclusions have been summarised.

Removed – page 2:

'The strongest emotion-specific effects were for happiness, with reduced accuracy ($p = .002$, $d_z = 0.49$) and sensitivity ($p = .004$, $\eta_p^2 = .17$) during heightened state anxiety.'

- 2. Furthermore, the "biases" should be phrased otherwise or explained within parentheses at some point of the abstract to clarify what is meant, e.g., "[state anxiety] appears to facilitate the processing of anger but impair the detection of happy expressions." Or something along those lines.**

We have now clarified the bias terms in the abstract.

Changed – page 2:

'High state anxiety reduced global emotion recognition accuracy ($p = .01$, $\eta_p^2 = .14$), but it did not affect interpretation bias towards perceiving anger in ambiguous angry–happy facial morphs ($p = .18$, $\eta_p^2 = .04$). We found no clear evidence of a relationship between trait anxiety and global emotion recognition accuracy ($p = .60$, $\eta_p^2 = .01$) or interpretation bias towards perceiving anger ($p = .83$, $\eta_p^2 = .001$). However, there was greater interpretation bias towards anger (i.e., away from happiness) during heightened state anxiety, among individuals with high trait anxiety ($p = .03$, $d_z = 0.33$). State anxiety appears to impair emotion recognition accuracy, and among individuals with high trait anxiety, it appears to increase biases towards perceiving anger (and away from happiness). Trait anxiety alone does not appear to be associated with facial emotion processing.'

- 3. The definition of trait and state anxiety deserves a sentence of its own, so this should be separated from the rest of the sentence: "are transient reactions to environmental stressors (31, 32)."**

The part of the sentence 'are transient reactions to environmental stressors' was part of the definition of state anxiety, but we have removed this if it was unclear.

Changed – page 5:

'Trait differences in anxiety exist between individuals and are more stable over time, whereas state variation in anxiety exists between individuals and within an individual over time (34, 35).'

- 4. Power calculations: the authors should report the power afforded by the current sample assuming a d of .69.**

We have reported this on page 16:

'We therefore had 87% power to detect our target effect size of $d_z = 0.69$.'

- 5. Internal consistency estimates (preferably McDonald's Omega) should be reported for all measures of interest, including both questionnaire and task-based measures.**

We have now reported internal consistency estimates for all questionnaires and tasks.

Added – page 18:

'Finally, we conducted *post hoc* internal consistency analyses of the measurements obtained from the questionnaires and tasks (Supplementary Table 3). McDonald's Omega (66) was used for the questionnaires, Cronbach's alpha was used for the 6AFC task, and the parallel forms reliability method was used for the 2AFC task because alternative methods (e.g., Cronbach's alpha) could not be applied to the 2AFC task (see the Supplementary Information for details). As shown in Supplementary Table 3, the measurements obtained from the questionnaires and the 6AFC task

(emotion recognition accuracy) had high internal consistency reliability, whereas the measurements obtained from the 2AFC task (interpretation bias towards perceiving anger) had moderate internal consistency reliability.'

Supplementary Table 3.

Estimates of Internal Consistency for the Tasks and the Questionnaires.

Measure	Low state anxiety	High state anxiety
Tasks		
Emotion recognition accuracy	.88	.80
Interpretation bias towards perceiving anger	.37 (.54)	.58 (.73)
Questionnaires		
STAI-state	.94	.94
PANAS-positive	.94	.88
PANAS-negative	.81	.86
VAS-positive	.75	.79
VAS-negative	.90	.93
Questionnaires (No experimental condition)		
STAI-trait	.94	
Anxiety sensitivity	.87	
Extraversion	.85	
Neuroticism	.89	
Lie	.80	

Emotion recognition accuracy was measured by global hits on the six-alternate forced choice (6AFC) task. Interpretation bias towards anger was measured by threshold scores on the two-alternate forced choice (2AFC) task. Cronbach's alpha was used for the 6AFC task. The parallel forms reliability method was used for the 2AFC task¹. We present Pearson's correlation coefficients (r) and Spearman-Brown corrections (ρ ; corrected for a full-length test) in brackets for the 2AFC task. McDonald's Omega was used for the questionnaires. Due to an error in the Hayes OMEGA macro, we were unable to calculate internal consistency for the psychoticism scale. STAI-state and STAI-trait = State-Trait Anxiety Inventory State Subscale and Trait Subscale, respectively; PANAS = Positive and Negative Affect Schedule; VAS = Visual Analogue Scale.

¹ Other methods of assessing internal consistency of the measurements obtained from tasks, such as Cronbach's alpha were not appropriate for the 2AFC task. This is because threshold scores (the outcome measure from the task) are calculated using a formula that accounts for the number of stimuli and trials (i.e., they are not simply a sum of responses to each trial). Consequently, correlating randomly selected sets of trials would be meaningless without ensuring that all 15 face images are included in equal proportions. We therefore split the 45 trials into three sets of 15 trials (i.e., each face image was shown once in each set), which represented parallel forms. Pearson's correlation coefficients and Spearman-Brown corrections were calculated for each of the three paired combinations of sets, and a mean average was taken.

An interpretation of these results has been added to the discussion:

Added – page 19:

'*Post hoc* analyses suggest that the measurements obtained from the 2AFC task (interpretation bias towards perceiving anger) had moderate reliability as measured by internal consistency. As stated by Parsons and colleagues (67), 'reliability is estimated from the scores obtained with a particular task performed by a particular sample under specific circumstances.' Therefore, although the task and the method of anxiety induction were consistent across the current and previous study, it is possible

that the samples differed in some way, which may account for this lack of replication. Nonetheless, there are no a priori reasons to suspect that this sample is systematically different from the previous study, and so the robustness of our earlier finding is questionable.'

6. All p-values should be reported to the third decimal place, following APA good practice recommendations.

RSOS is not an APA journal, and we understand that the Royal Society supports format-free initial submission (<https://royalsociety.org/journals/authors/author-guidelines/#formatting>). We have reported p-values to 2 decimal places as we consider this level of precision to be sufficient. However, where a number is below .01, we have reported it to 3 decimal places (e.g., .004), as at least one significant digit should be reported (<https://www.ncbi.nlm.nih.gov/pmc/articles/PMC4483789/>).

7. I am missing a discussion on the potential behavioral consequences of the observed biases. Without straying too far from the data, the authors should briefly discuss how the observed biases might play out in terms of real-life approach-avoidance behavior. This would also dovetail with the Introduction, where the authors do comment on these issues. Moreover, the authors should more precisely delimit the relevance of the reported effects for anxiety in comparison to other psychopathological dimensions, as similar biases have been reported for individuals high on trait anger and aggression.

We have expanded on the potential behavioural consequences of the biases in the discussion.

Added – page 23:

'This could lead to inappropriate or blunted reactions during social interactions or behavioural avoidance, which may evoke negative reactions from others (13), thus potentially impacting attachments and relationships.'

Regarding your second point, although there is some evidence that individuals high in self-reported aggression are more likely to misidentify anger in facial cues (Hall, 2006). Research from our group using the same behavioural tasks (6AFC and 2AFC) has found no clear evidence of a main effect of trait aggression on global emotion recognition accuracy (total hits) or interpretation bias towards anger (Eastwood et al., 2020). Therefore, we would argue that our effects for state anxiety are not similar to those for aggression on the tasks employed here.

<https://psycnet.apa.org/record/2006-11593-007>

<https://journals.sagepub.com/doi/full/10.1177/0269881120922951>

Reviewer 4:

Abstract:

1. As reviewer 1 already mentioned, it is unusual to report statistics in such an amount in the abstract as it decreases readability. If you really want such information in the abstract, please be consistent and report effect sizes to all p-values and use either two or three decimal places

We understand your point about readability. However, in our experience it is more informative and transparent to report statistics in the abstract to support results statements. Therefore, we would

prefer to keep this information. We have now added the Cohen's d_z effect size that was missing, for consistency.

Added – page 2:

'However, there was greater interpretation bias towards anger (i.e., away from happiness) during heightened state anxiety, among individuals with high trait anxiety ($p = .03$, $d_z = 0.33$).'

We have reported p-values to 2 decimal places as we consider this level of precision to be sufficient. However, where a number is below .01, we have reported it to 3 decimal places (e.g., .004), as at least one significant digit should be reported

(<https://www.ncbi.nlm.nih.gov/pmc/articles/PMC4483789/>).

2. Reporting the place of recruitment in the abstract is also highly unusual and doesn't add valuable information in my opinion

We think that geographic location (country) of the study adds context. Furthermore, a reviewer previously asked us to report where the sample was recruited from. However, we have now made this more concise by omitting the city, as we agree that this is superfluous.

Changed – page 2:

'... in a laboratory experiment with healthy UK participants...'

3. I don't think that "7.5% carbon dioxide challenge" is a widely known paradigm, maybe you can enter a short descriptive sentence about this experimental manipulation including its sham equivalent instead

Added – page 2:

'High and low state anxiety were induced via inhalations of 7.5% carbon dioxide enriched air and medical air, respectively.'

Introduction:

4. Last paragraph p4: "Measurement differences may also explain mixed findings. In general, accuracy reflects hits (correct identification of emotions), bias reflects hits and false alarms (incorrect identification of emotions), and sensitivity reflects hits whilst accounting for false alarms." ☒ Statistic might be a better word compared to measurement. This whole paragraph is quiet confusing and demands a lot of inference by the reader. Please clarify your arguments.

This paragraph has been expanded on to clarify the differences in the definitions of these terms and their interconnections.

Changed – page 5:

'In short, emotion recognition accuracy may be measured by hit rate (e.g., the correct identification of anger if angry faces are presented. If an individual demonstrates a higher hit rate for anger, this suggests that they have superior emotion recognition accuracy for anger. However, this measure of emotion recognition accuracy does not account for the times when an individual identifies anger in faces that are not angry). These are known as false alarms/errors (i.e., the incorrect identification of the emotions presented). A bias towards making angry responses may manifest in a higher hit rate and a higher false alarm rate, whereas sensitivity reflects hit rate whilst accounting for false alarms.

An additional complication is that the term ‘bias’ can also refer to different things (e.g., neutral or ambiguous emotional facial expressions interpreted as angry) depending on the task and stimuli used in a study. Furthermore, different statistical analyses can be used to measure both bias and sensitivity.’

5. P5L5: What is “naturally occurring state Anxiety”?

Clarified – page 5:

‘...naturally occurring (i.e., not experimentally manipulated) state anxiety...’

6. P5L18: What is “7.5% carbon dioxide (CO₂) challenge”? Maybe you don’t have to define this paradigm here, but can describe the conceptual idea “participants inhale XXX to XXX “

Added – page 6:

‘This experimental manipulation compares the effects of a 20-minute inhalation of 7.5% CO₂ enriched air versus a 20-minute inhalation of medical air (control), while tasks are performed.’

Method:

7. P7L39: “Participants were eligible if they met our criteria for low or high trait anxiety” Which are? Please refer to the paragraph where you describe your criteria

Added – page 8:

‘...(details in the following paragraph)...’

8. P8L10: “Scores between 21-31 and 44-64 denoted low and high trait anxiety, respectively. For the last two participants, we lowered the threshold of inclusion for high trait anxiety to 41-64 to support recruitment.” Very complicated, maybe only: “Scores between 21-31 and 41-64 denoted low and high”?

As a number of steps were taken to reach these criteria, as outlined in the paragraph, we think it is important to report these details for transparency. We have now changed this sentence and added the extra details to a footnote to aid readability.

Changed – page 9:

‘Scores between 21-31 and 41-64 denoted low and high trait anxiety, respectively¹.’

¹ ‘Scores between 21-31 and 44-64 denoted low and high trait anxiety, respectively for the majority of participants. For the last two participants, we lowered the threshold of inclusion for high trait anxiety to 41-64 to support recruitment.’

9. P9L26: Maybe you like to add the following reference which found that neutral and angry faces elicit comparable negative facial responses when they are passively viewed which indicates the negative valence of neutral faces: Höfling, T., Alpers, G. W., Gerdes, A. B. M., & Föhl, U. (2021). Automatic Facial Coding Versus Electromyography of Mimicked, Passive and Inhibited Facial Response to Emotional Faces. *Cognition and Emotion*, 35(5), 874-889. <https://doi.org/10.1080/02699931.2021.1902786>

Added – page 10:

'For example, neutral and angry faces have been found to elicit comparable negative facial responses when passively viewed, which may indicate that neutral faces are perceived to be negatively valenced (58).'

10. P10L12: What is "A-prime"?

We defined A-prime on page 15:

'A' is a distribution free, non-parametric, signal detection measure of sensitivity (62). We calculated sensitivity scores from hit rate and false alarm rate data for each emotion in each state anxiety condition, based on the formula provided by Fisk and colleagues (64). For a small number of cases where hits were less than false alarms, we used an alternative formula (62), verified using software (65). A' sensitivity scores typically range from .5 (signal cannot be distinguished from noise) to 1 (perfect performance), values less than .5 may arise from sampling error or response confusion, and the minimum possible value is 0 (62).'

11. ANOVA is a common abbreviation

By this comment, we presume the reviewer means that our expansion on first use is unnecessary. In line with RSOS author guidelines, we have not included any unexplained abbreviations or acronyms. Although ANOVA is a common abbreviation, we want to avoid any ambiguity for readers. We are happy to defer to the journal style here.

Results:

12. Again, varying decimal place of p-values, please unify

We have reported p-values to 2 decimal places as we consider this level of precision to be sufficient. However, where a number is below .01, we have reported it to 3 decimal places (e.g., .004), as at least one significant digit should be reported (<https://www.ncbi.nlm.nih.gov/pmc/articles/PMC4483789/>).

13. "There was also strong evidence of a state anxiety emotion interaction [$F(3.76, 165.36) = 4.11, p = .004, \eta^2 = .09$]. ☒ This is surely not a large effect, please indicate your effect size interpretation in the corresponding method section

Although the effect is not large, the evidence for it is strong. Nonetheless, we have removed the word 'strong' from this sentence, and we have added information to the methods section to describe how we have interpreted the evidence.

Added – page 14:

'Results are framed in terms of the strength of evidence against the null hypothesis (e.g., $p < .05$ provides modest evidence whilst $p < .001$ provides strong evidence) (60). Cohen (61) has also provided conventions to define small ($\eta^2 = .01; dz = 0.20$), medium ($\eta^2 = .06; dz = 0.50$), and large ($\eta^2 = .14; dz = 0.80$) effects.'

14. The assumption of sphericity was violated for the emotion, and state anxiety $\hat{A} \sim$ emotion within-subjects effects, so here we report Greenhouse Geisser statistics. ☒ Such information needs to be in the corresponding method section and not in the results section

Added – page 13:

'Where the assumption of sphericity was violated, Greenhouse Geisser statistics are reported.'

Changed – page 16:

‘Greenhouse Geisser corrections were applied to both effects.’

15. P16L33: non-significant differences of mean scores are interpreted. There is clearly no difference between them.

We had included this information in the manuscript because we originally thought that it was appropriate to comment on the direction of the differences for findings related to a directional hypothesis. However, to improve clarity, we have removed these.

Removed – page 17:

‘Although scores were lower in the high state anxiety ($M = 6.78$, $SD = 1.41$) than the low state anxiety ($M = 7.07$, $SD = 1.14$) condition, there was no clear statistical evidence of a difference.’

Removed – page 19:

‘However, results were in the direction we predicted; there was an increased tendency to perceive anger and a decreased tendency to perceive happiness in the high (versus low) state anxiety condition.’

16. Please move your manipulation check from the end to the beginning of the results (order of tables is currently wrong too)

The manipulation check paragraph has been moved to near the start of the results section (after participant characteristics). Because baseline participant characteristics and results of the manipulation check are presented in Table 1, we think this addresses the point about table order. All tables are presented and referred to in the correct order.

17. “gas inhalation order modified the effect of state anxiety on emotion recognition accuracy ($p = .01$), with stronger effects when the 7.5% CO₂ inhalation came first.” ☐ Is the robustness of your findings threatened? Please provide statistics and more details on this effect as a supplement

No, we do not think the robustness of our findings are threatened. This is a procedural issue and a common finding in CO₂ inhalation studies. In our experience, we have generally found that the main effect of gas inhalation on the outcome is stronger when the CO₂ inhalation comes first. This is likely to reflect a stronger ‘dose’ of anxiety in this order group, as participants experience the anticipatory anxiety of the procedure as well as CO₂ induced anxiety. Furthermore, we find that the effect of either gas inhalation (air or CO₂) on the outcome is generally stronger when it comes first rather than second. Participants who receive the air inhalation first do have the anticipatory anxiety, so a weakened anxiety signal is still present in the ‘control’ condition when the gases are given in this order.

Added – page 18:

‘This is a common finding in CO₂ inhalation studies, reflecting a possible effect of anticipatory anxiety in addition to the CO₂ induced anxiety.’

We have now added a table of these results to the supplementary materials.

Supplementary Table 2

Moderation Effects of Gas Inhalation Order and Task Order on the Effects of State and Trait Anxiety on Interpretation Bias towards Perceiving Anger (2AFC) and Emotion Recognition Accuracy (6AFC)

	Gas inhalation order		Task order	
	F	p-value	F	p-value
Interpretation bias towards perceiving anger (2AFC)				
State anxiety	0.15	.70	1.39	.25
Trait anxiety	1.16	.29	0.05	.83
Emotion recognition accuracy (6AFC)				
State anxiety	7.84	.01	0.03	.86
Trait anxiety	2.42	.13	2.58	.12

Note: 2AFC = two-alternate forced choice task and 6AFC = six-alternate forced choice task. Degrees of freedom = 1,38. Stratified results for the moderation effect of gas inhalation order showed that the main effect of state anxiety on emotion recognition accuracy was stronger when the 7.5% CO₂ inhalation came first [$F(1, 21) = 11.84, p = .002$], compared to when the air inhalation came first [$F(1,21) = 0.05, p = .82$].